# Circulation timescales of Atlantic Water in the Arctic Ocean determined from anthropogenic radionuclides

Anne-Marie Wefing[1,2], Núria Casacuberta[1,2], Marcus Christl[1], Nicolas Gruber[2], and John N. Smith[3]

[1]Laboratory of Ion Beam Physics, Institute for Particle Physics and Astrophysics, ETH Zürich, Switzerland
[2]Environmental Physics, Institute of Biogeochemistry and Pollutant Dynamics, ETH Zürich, Switzerland
[3]Bedford Institute of Oceanography, Fisheries and Oceans Canada, Dartmouth, NS, Canada

**Correspondence:** Anne-Marie Wefing (awefing@phys.ethz.ch)

**Abstract.** The inflow of Atlantic Water to the Arctic Ocean is a crucial determinant for the future trajectory of this ocean basin with regard to warming, loss of sea-ice and ocean acidification. Yet many details of the fate and circulation of these waters within the Arctic remain unclear. Here, we use the two long-lived anthropogenic radionuclides $^{129}$I and $^{236}$U together with two age models to constrain the pathways and circulation times of Atlantic Water in the surface ($10-35\,\mathrm{m}$ depth) and in the mid-depth Atlantic layer ($250-800\,\mathrm{m}$ depth). We thereby benefit from the unique time-dependent tagging of Atlantic Water by these two isotopes. In the surface layer, a binary mixing model yields tracer ages of Atlantic Water between 9-16 years in the Amundsen Basin, 12-17 years in the Fram Strait (East Greenland Current) and up to 20 years in the Canada Basin, reflecting the pathways of Atlantic Water through the Arctic and their exiting through Fram Strait. In the mid-depth Atlantic layer ($250-800\,\mathrm{m}$), the transit time distribution (TTD) model yields mean ages in the central Arctic ranging between 15 and 55 years, while the mode ages representing the most probable ages of the TTD range between 3 and 30 years. The estimated mean ages are overall in good agreement with previous studies using artificial radionuclides or ventilation tracers. Although we find the overall flow to be dominated by advection, the shift of the mode age towards a younger age compared to the mean age reflects also the presence of a substantial amount of lateral mixing. For applications interested in how fast signals are transported into the Arctic's interior, the mode age appears to be a suitable measure. The short mode ages obtained in this study suggest that changes in the properties of Atlantic Water will quickly spread through the Arctic Ocean and can lead to relatively rapid changes throughout the upper water column in future years.

## 1 Introduction

### 1.1 The role of Atlantic Water in the Arctic Ocean

The Arctic sea-ice extent in 2019 was one of the lowest in the satellite record but in line with the long-term trend of declining ice cover (e.g., "Special Report on the Ocean and Cryosphere in a Changing Climate (IPCC)", 2019). In addition to the main contribution coming from the solar heating of the surface mixed layer in summer (Carmack et al., 2015), recent works highlighted also the contribution of the inflow of warm Atlantic Water (AW) through Fram Strait to the observed sea-ice

decrease (Polyakov et al., 2005, 2017; Årthun et al., 2019). This is especially relevant for the Eurasian Basin of the Arctic Ocean and the process is now commonly known as "Atlantification".

AW constitutes about $90\%$ of the total inflow of waters to the Arctic Ocean with the remaining composed of $\sim 9\%$ of Pacific Water entering through Bering Strait and $\sim 1\%$ of freshwater that includes runoff and precipitation (e.g., Woodgate, 2013). Both Pacific Water and freshwater largely reside in the upper water column of the Arctic Ocean, including the Polar Mixed Layer and the upper halocline (hereafter referred to as surface layer). AW enters the Arctic Ocean through either the Fram Strait or the Barents Sea, forming the Fram Strait Branch Water (FSBW) and Barents Sea Branch Water (BSBW), respectively. These

two branches encounter one another in the St. Anna Trough and constitute the mid-depth Atlantic layer of the Arctic Ocean, also referred to as Arctic Atlantic Water (AAW), which is found between 250 and $800\,\mathrm{m}$ depth. This water mass is characterized by higher temperatures and densities compared to the overlying halocline waters (e.g., Rudels, 2015). Core FSBW is generally found in about $300-400\,\mathrm{m}$ depth and can be associated with a potential density of $\sigma_\Theta = 27.91$. The BSBW core characterized by $\sigma_\Theta = 28$ is located below, at about $800\,\mathrm{m}$ depth (Karcher et al., 2012). Both branches circulate through the Arctic Ocean

mainly under a cyclonic regime (Fig. 1), following the pathways of the Arctic Ocean Boundary Current (Mauldin et al., 2010; Rudels et al., 2012). However, this pattern might be subject to temporal variations (Karcher et al., 2012).

Despite ongoing work, pathways of AW circulation in the Arctic Ocean are still not well understood. Generally, AW can be traced by its T-S properties, atmospherically introduced anthropogenic pollutants such as CFCs, or anthropogenic radionuclides introduced from atmospheric nuclear weapon tests as well as nuclear reprocessing plants (Woodgate, 2013). AW circulation

times or water mass ages can only be obtained from transient tracers and are rather poorly constrained. Recent studies suggest circulation times on the order of 15-30 years from the Barents Sea opening through the Arctic Ocean and back to the Fram Strait (Smith et al., 2011; Karcher et al., 2011).

As a consequence of ongoing climate change, increased freshwater input from melting sea ice as well as changes in the Atlantic and Pacific inflows may lead to changes in the circulation pattern in future years (e.g., Woodgate, 2013). Acquiring a

better understanding of circulation pathways and their temporal variations will be key to predicting the nature of future changes in the Arctic domain, examples being the propagation of increased AW temperature or the accumulation of anthropogenic $CO_2$ (Terhaar et al., 2020a). Therefore, suitable tracers are required that unambiguously label AW in the Arctic Ocean and at the same time provide information about timescales associated with the circulation of AW to revise present circulation times and trace future changes.

In this study, we will investigate circulation pathways, mixing regimes as well as tracer ages of AW in the surface and Atlantic layers of the Arctic Ocean. This will be done using a novel approach that combines the two long-lived anthropogenic radionuclides $^{129}$I and $^{236}$U. Obtained ages will be put into the context of available literature data and different approaches on how to estimate circulation timescales will be compared. Strengths and weaknesses especially of applying the transit time distribution model introduced in section 1.3 to anthropogenic radionuclides will be discussed and implications for the Arctic

Ocean will be highlighted.

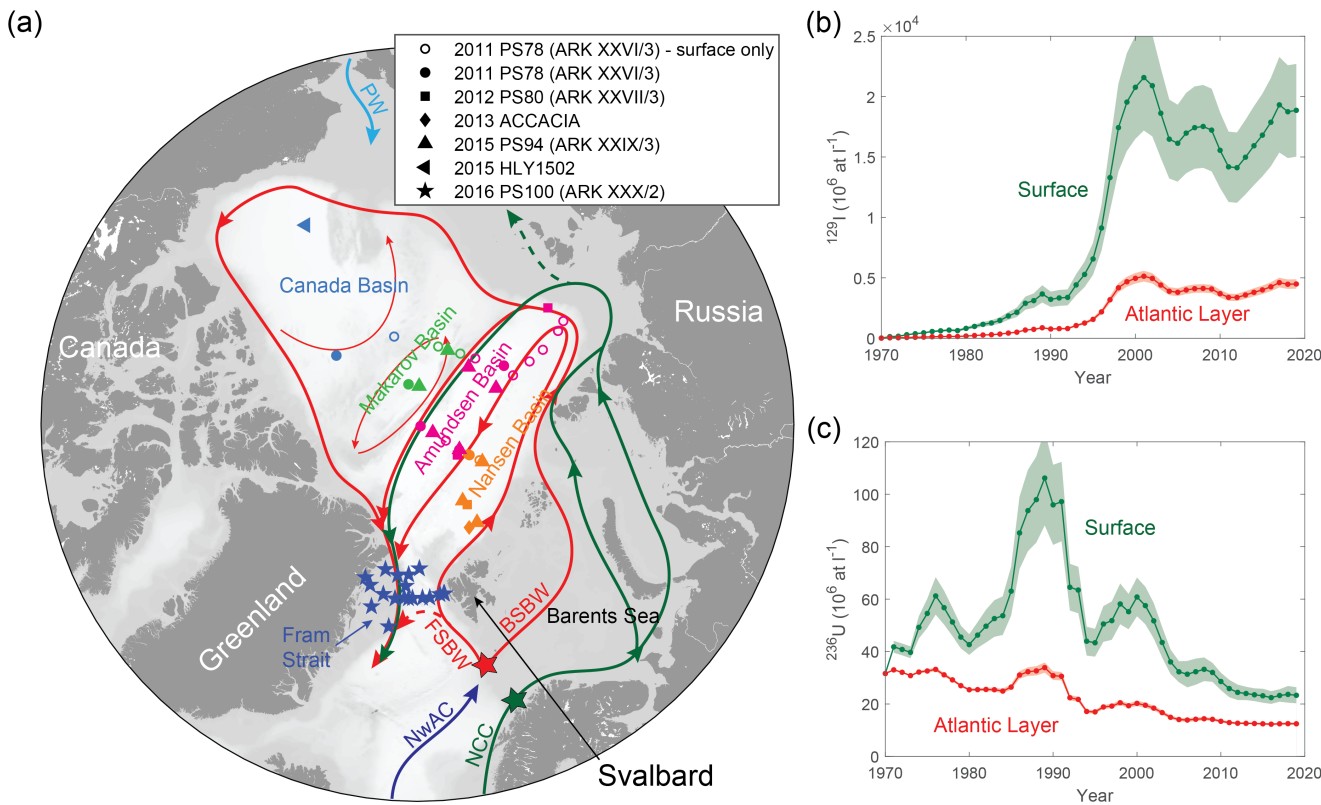

**Figure 1.** (a) Circulation of Atlantic Water in the surface (green) and the Atlantic layer (red) of the Arctic Ocean. Ocean currents are Norwegian Atlantic Current (NwAC, dark blue), Norwegian Coastal Current (NCC), Fram Strait Branch Water (FSBW), Barents Sea Branch Water (BSBW) and Pacific Water (PW, light blue). Symbols show stations where samples for $^{129}$I and $^{236}$U analysis were taken during different expeditions between 2011 and 2016 (see figure legend). Different colors correspond to different Arctic basins. Red and green star symbols mark the locations for which the input functions shown in (b) and (c) are defined. (b) $^{129}$I input functions for the surface (green) and Atlantic layer (red). (c) $^{236}$U input functions for the surface (green) and Atlantic layer (red). Uncertainties were propagated from uncertainties in reprocessing plant Uranium releases (Christl et al., 2015) and fractions of La Hague, Sellafield, Global Fallout (Casacuberta et al., 2018).

## 1.2 $^{129}$I and $^{236}$U as tracers of Atlantic Water in the Arctic Ocean

Several recent studies have employed the combination of two long-lived anthropogenic radionuclides, $^{129}$I and $^{236}$U, as tracers for the circulation of AW in the Arctic Ocean and Fram Strait (Casacuberta et al., 2016, 2018; Wefing et al., 2019). In contrast to other established anthropogenic radionuclide tracers such as $^{137}$Cs, the half-life ($t_{1/2}$) of $^{236}$U is around 23 million years and therefore of the same order of magnitude as for $^{129}$I with $t_{1/2} = 16$ million years. Thus, these two isotopes can be considered as conservative tracers whose application as age tracers comes from their strongly time-dependent input function (Casacuberta

et al., 2018). An important advantage of the two isotopes is their low sampling volume. Measurements of $^{129}$I and $^{236}$U require only around $250\,\mathrm{ml}$ and $3-5\,\mathrm{l}$ of seawater, respectively.

$^{129}$I and $^{236}$U have been introduced to the marine environment from two sources: (i) Liquid releases from nuclear fuel reprocessing plants (mainly $^{129}$I) and (ii) Global fallout from atmospheric nuclear weapon tests (mainly $^{236}$U) (Casacuberta et al., 2016). In the Arctic Ocean, releases from the two European reprocessing plants, the Sellafield nuclear reprocessing plant, United Kingdom, and the La Hague nuclear fuel reprocessing plant, France, are the dominant source of $^{129}$I and $^{236}$U followed by the global fallout signal. The natural background of both radionuclides in the Arctic Ocean is several orders of magnitude lower compared to the inputs from anthropogenic activities (Casacuberta et al., 2016).

The liquid releases from the two European reprocessing plants are transported to the Arctic Ocean mainly by the Norwegian Coastal Current (NCC) (Christl et al., 2015; Edmonds et al., 2001; Gascard et al., 2004), which supplies the Arctic shelf seas and the surface layer of the Eurasian Basin (Rudels (2015), Fig. 1a, dark green). After their circulation through the Arctic Ocean, waters from the surface layer largely exit it via the Fram Strait, forming the upper layer of the East Greenland Current.

The much smaller global fallout signal is carried into the Arctic Ocean by all currents. This includes the input of Pacific Water through the Bering Strait, which remains in the upper water column and dominates the water properties in the Canada and Makarov basins. The input from the Atlantic occurs primarily through the Norwegian Atlantic Current (NwAC). This current carries also a part of the reprocessing plant signal as a result of the partial mixing with the NCC that brings this signal northward from the reprocessing plants. Although only a small part of the NCC is actually mixed with the NwAC resulting in a strongly diluted signal, the radioisotope signal carried by the NwAC is still much larger compared to the pure global fallout signal carried by Pacific Water. This mixture then enters the Arctic Ocean via the FSBW and the BSBW (Casacuberta et al. (2018), Fig. 1a, red), supplying the mid-depth waters. Thus, AW is uniquely tagged with the reprocessing plant signal of $^{129}$I and $^{236}$U, permitting us to assess the pathways and circulation times of AW in the Arctic Ocean.

## 1.3 Using $^{129}$I and $^{236}$U as transient tracers to estimate circulation times

The quantitative use of $^{129}$I and $^{236}$U as transient tracers for the estimation of AW circulation times or water mass ages requires accurate knowledge of the tracer input functions. These input functions need to be constructed for the location where the tracer signal enters the study area. Consequently, this is the location where the tracer age is defined as zero. Different models can then be applied to determine the water mass age from the tracer measurements, such as a binary mixing or a transit time distribution (TTD) model. The former has been used with anthropogenic radionuclides in earlier studies (e.g., Smith et al., 1998, 2005, 2011; Christl et al., 2015; Wefing et al., 2019), where it has also been referred to as a "tracer age model" or "dual-tracer approach". The TTD model has been applied widely in the context of ocean interior ventilation studies (e.g., Haine and Hall, 2002; Waugh et al., 2003; Smith et al., 2011; Stöven et al., 2015; Tanhua et al., 2009), and to determine the oceanic uptake of anthropogenic $CO_2$ (e.g., Hall et al., 2002; Waugh et al., 2006; Tanhua et al., 2008; Khatiwala et al., 2009; Olsen et al., 2010; Khatiwala et al., 2013; Stöven and Tanhua, 2014; Stöven et al., 2016; He et al., 2018). In this study the binary mixing model will be applied to samples from the surface layer, whereas the TTD model will be used for the mid-depth Atlantic layer due to the following model characteristics:

In the binary mixing model, only two processes are considered. The first one is lateral advection that carries the tracer signal defined in the input function into the Arctic Ocean and out again without lateral mixing. The second one is mixing with water masses that contain a constant (low) background radionuclide concentration from global fallout (here, e.g., Pacific Water). The relationship between the concentrations of two tracers then permits a definition of a dilution factor and a single tracer age for each water parcel (Smith et al., 1998). The assumptions behind the binary mixing model are reasonable for the surface layer of the Arctic Ocean, as shown in Smith et al. (2011), given its strong lateral confinement. However, these assumptions generally do not hold for the mid-depth Atlantic layer and for most other parts of the world's oceans, since lateral mixing by, e.g., mesoscale processes is a strong feature of the ocean's flow.

In the TTD method this lateral mixing is explicitly considered in addition to the purely advective flow. This leads to a probability density function (PDF) of water mass ages, or a distribution of transit times. The TTD method is an inversion method that aims to solve for the flow characteristics of an advection-diffusion problem given information about the distribution of tracers containing time information. In order for this problem to remain solvable, the flow problem is often limited to 1D, and the boundary conditions at the source have to be well known. This is typically given for atmospherically introduced transient tracers such as CFCs, $SF_6$ or $^3H$ with known input functions, or natural radionuclides such as $^{39}Ar$ or $^{14}C$ that have relatively constant surface concentrations and are only subject to radioactive decay. An important limitation of the TTD method is that it is not straightforward to extend it to the consideration of the mixing of different endmembers, especially when the endmembers are hard to "unmix" (Haine and Hall, 2002). In the Arctic Ocean, only the mid-depth Atlantic layer can be considered to consist of one endmember since no mixing with Pacific Water is assumed to take place, justifying our use of a one endmember TTD model (e.g., Rudels, 2015). This is not the case for the surface layer hence only the binary mixing model will be applied there.

To our knowledge only one study has applied the TTD model to determine lateral transit times of AW in the Arctic Ocean using anthropogenic radionuclides (Smith et al., 2011). These authors combined radionuclides from nuclear reprocessing ($^{129}I$ and $^{137}Cs$) with the atmospherically introduced tracer CFC-11. By introducing those radionuclides with a point-like source function in addition to the classically used CFCs, Smith et al. (2011) substantially expanded the application of the TTD to the lateral flow of AW. The combination of these tracers could be applied to the mid-depth Atlantic layer because the AW is largely isolated from the atmosphere in the eastern Norwegian Sea, proximal to the region where the radionuclides' input function is defined. Hence the input functions for both sets of tracers could be initialized at the same location. This permits the comparison of lateral circulation ages from anthropogenic radionuclides with CFC-derived ventilation ages.

In this study the two models presented above will be applied to revise circulation features and investigate circulation times of AW in the Arctic Ocean using the combination of $^{129}I$ and $^{236}U$. The detailed methodological approach is outlined in the following section.

**Table 1.** Water masses and properties in the two layers that samples were clustered by in this study, following the definitions in Rudels et al. (2005). PSW: Polar Surface Water, AAW: Arctic Atlantic Water, AIW: Arctic Intermediate Water

| Layer | Depth range (m) | Water mass | $\Theta$ (°C) | Pot. Density $\sigma_\Theta$ |
|---|---|---|---|---|
| Surface layer | $10 - 35\,\mathrm{m}$ | PSW | $< 0$ | $\leq 27.7$ |
| | | Warm surface water | $> 0$ | $\leq 27.7$ |
| Atlantic layer | $250 - 800\,\mathrm{m}$ | AAW | $0 - 2$ | $27.7 - 27.97$ or $> 27.7, \sigma_{0.5} \leq 30.444$ |
| | | AIW | $< 0$ | $> 27.7$ |

## 2 Materials and Methods

### 2.1 Data

This study is based on $^{129}$I and $^{236}$U data from various expeditions in the Arctic Ocean and Fram Strait between 2011 and 2016 (Fig. 1 and Table S1). All seawater samples were collected from Niskin bottles during CTD casts, chemically purified and measured using Accelerator Mass Spectrometry (AMS). Details on sample collection and chemical treatment can be found in Casacuberta et al. (2016, datasets PS78 2011, PS80 2012), Casacuberta et al. (2018, dataset PS94 2015) and Wefing et al. (2019, dataset PS100 2016). Samples from the Nansen Basin in 2013 (ACCACIA) were also treated according to the protocols in Wefing et al. (2019). All these samples were measured at the Laboratory of Ion Beam Physics at ETH Zurich. Samples collected in the Canada Basin in 2015 (HLY1502) were processed and measured at the A. E. Lalonde AMS Laboratory at the University of Ottawa ($^{129}$I) and the Centro Nacional de Aceleradores in Sevilla and ETH Zurich ($^{236}$U). For the scope of this study, we only considered samples from the surface layer and the mid-depth Atlantic layer as defined in Table 1.

### 2.2 $^{129}$I and $^{236}$U input functions

Two different input functions were defined for $^{129}$I and $^{236}$U, respectively: one for the surface and one for the mid-depth Atlantic layer (Fig. 1b and c). Both input functions were reconstructed from 1901 to 2019 in the following way: for 1901 to 1949 only the natural background was considered, which was assumed to be zero for both isotopes. From 1950 to 1971, only the global fallout tracer signal was considered and from 1971 to 2019, the signals from global fallout and from reprocessing plant releases were combined.

For the global fallout signal we assumed a temporally constant $^{129}$I concentration of $1 \times 10^7\,\mathrm{at \cdot l^{-1}}$ but used a time-dependent $^{236}$U concentration. This is valid because the atmospheric weapon tests in the 1950s and 60s introduced significantly less $^{129}$I compared to $^{236}$U (Raisbeck and Yiou, 1999; Sakaguchi et al., 2009). The $^{236}$U global fallout signal was calculated from a diffusive global surface ocean model with $1° \times 1°$ lateral resolution fed by the latitudinally averaged atmospheric depositional flux of $^{236}$U from weapon tests (Christl et al., 2015). To determine the relevant concentration for our tracer input function for the Arctic Ocean we used the model output for $0 - 50\,\mathrm{m}$ depth at a location of $74°\,\mathrm{N}$ and $19°\,\mathrm{E}$. This location corresponds

approximately to the region where the NwAC bifurcates into FSBW and BSBW (Fig. 1a). This is also the region where the reprocessing plant signal carried by the NCC is admixed to the NwAC. The modeled global fallout $^{236}$U concentration decreases from about $3.5 \times 10^7 \, \text{at} \cdot \text{l}^{-1}$ in the 1960s to about $1 \times 10^7 \, \text{at} \cdot \text{l}^{-1}$ in the early 1990s. For the input function after 1990 we used a fixed $^{236}$U concentration of $1 \times 10^7 \, \text{at} \cdot \text{l}^{-1}$, which matches recent measurements in the North Pacific Ocean (Eigl et al., 2017).

For the inputs of $^{129}$I and $^{236}$U from the two reprocessing plants we used the respective reported or reconstructed radionuclide releases (new data to 2019 were kindly provided by ORANO/IRSN, pers. comm., 2018) to model $^{129}$I and $^{236}$U in the streams emanating from the two sites, following the approach of Christl et al. (2015). In the present study the two streams were treated separately and for each branch, a three-year moving average was applied to account for mixing and water mass residence time in the North Sea. To combine the two streams and the global fallout signal into input functions for the Arctic Ocean, we adapted the approach of Casacuberta et al. (2018). They used measurements of $^{129}$I and $^{236}$U concentrations in the Fram Strait and Barents Sea in 2015 to determine the fraction of radionuclides coming from La Hague, Sellafield and global fallout in each of the three branches entering the Arctic Ocean, i.e., NCC (referred to as Arctic Shelf Break Branch therein) and the FSBW and BSBW, the latter two formed from the NwAC. In this study we used the fractions identified for the NCC for the surface layer input function and the average of the fractions identified for FSBW and BSBW for the Atlantic layer input function. Note that the surface and Atlantic layer input functions of $^{129}$I are in reasonable agreement ($\pm 10\%$) with those used by Smith et al. (2011) (Fig. S1). This proves the solidity of the input function as the two studies used a different approach combining the reprocessing plant releases and also defined the surface input function at a different location. The calculated $^{129}$I and $^{236}$U input functions (Fig. 1b and c, Table S2) were then used in the binary mixing model and the TTD model to investigate circulation times.

## 2.3 Models to determine circulation timescales and mixing regimes

### 2.3.1 Binary mixing model

In our application of the binary mixing model for the surface layer of the Arctic Ocean, we assumed that water labeled with the time-dependent tracer signal entering the Arctic Ocean in the surface AW input function only mixes with water carrying the global fallout signal of $1 \times 10^7 \, \text{at} \cdot \text{l}^{-1}$ for $^{129}$I and $^{236}$U (i.e., Pacific and Atlantic Water not labeled with reprocessing plant releases).

In order to estimate circulation times or tracer ages (both terms have been used synonymously), mixing lines between each year of the input function and the global fallout endmember were constructed in $^{236}$U vs. $^{129}$I tracer space (Fig. 2a). The tracer age, as well as the corresponding dilution factor (DF) of the input function, were constrained by plotting the measured data in the same tracer space. It is important to note that dilution here corresponds to a dilution with global fallout background. Other dilutions, such as the addition of waters from sea-ice melt and meteoric waters in the surface layer, were not considered. Those contain a negligible radionuclide signal (Casacuberta et al., 2016, 2018) and therefore would represent a third endmember in the $^{129}$I and $^{236}$U tracer space, albeit one that is close to the global fallout endmember. Thus, even in the presence of a substantial input by freshwater, the effect would be small, justifying our neglect of this input.

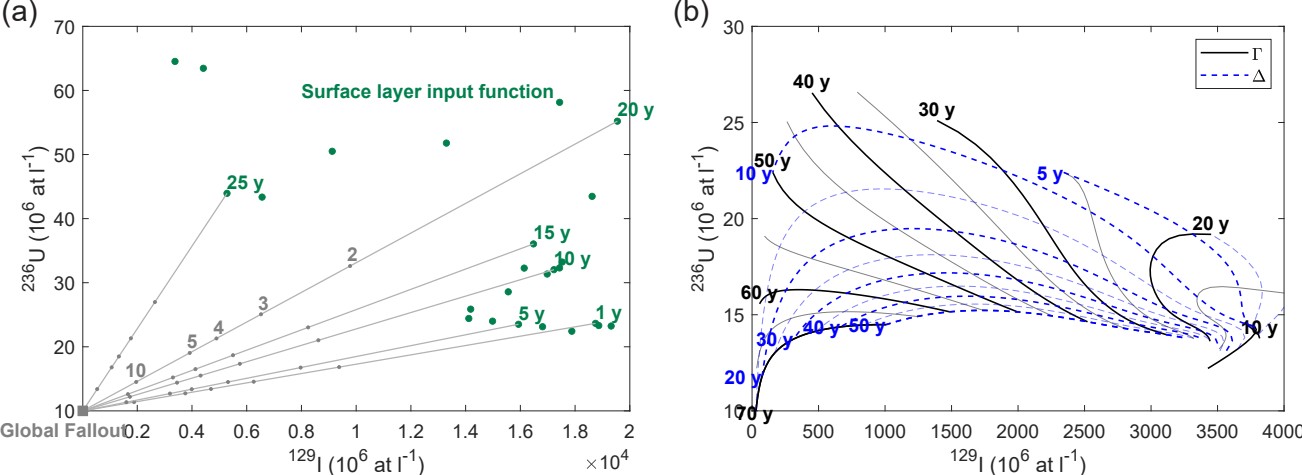

**Figure 2.** (a) Concept of the binary mixing model with a constant global fallout endmember (gray square) and a time-dependent surface layer input function (green dots with corresponding tracer ages, based on a collection year of 2019) in $^{236}$U vs. $^{129}$I tracer space. Tracer age curves converge towards the global fallout endmember at $1 \times 10^7$ at $\cdot$ l$^{-1}$ for $^{129}$I and $^{236}$U. Gray numbers denote different dilution factors of the input function. (b) Concept of the TTD model with $\Gamma$ (black) and $\Delta$ (blue) isolines in $^{236}$U vs. $^{129}$I tracer space, calculated from the Atlantic layer input functions for 2019. Numbers denote $\Gamma$ (mean age, black) and $\Delta$ (width of age distribution, blue) values in years.

The mixing lines converge towards the constant global fallout endmember, hence the determination of a tracer age becomes less precise for a strong dilution of tracer-labeled water with water labeled with only the global fallout signal. As the concept of the binary mixing model is based on the assumption of purely advective flow, tracer signals from different years are assumed not to mix with one another, i.e., only mixing between the input function signal from a single year with the constant global fallout signal is considered.

### 2.3.2 Transit time distribution model

In the TTD model that we applied to the study of the mid-depth Atlantic layer, the concentration of a stable tracer at sampling location $x$ and time $t$ is described by

$$c(x,t) = \int_0^\infty c_0\left(t - t'\right) G\left(x, t'\right) dt'. \tag{1}$$

The symbol $c_0(t)$ represents the tracer input function and $G(x,t)$ is the Green's function which propagates the tracer signal, and therefore describes the properties of the flow. One can consider $G(x,t)$ as the PDF that weighs the tracer signals from different years of the input function for a certain sampling time $t$ at location $x$. For one-dimensional transport described by the 1D advection-diffusion equation:

$$\frac{\partial c}{\partial t} + u\frac{\partial c}{\partial x} - k\frac{\partial^2 c}{\partial x^2} = 0, \tag{2}$$

(tracer concentration $c$, velocity $u$, diffusivity $k$), the PDF has the shape of an inverse Gaussian function:

$$G(x,t) = \frac{x}{\sqrt{4\pi k t^3}} \, exp\left(-\frac{(ut-x)^2}{4kt}\right). \tag{3}$$

Substituting $\Gamma = \frac{x}{u}$ and $\Delta = \sqrt{\frac{kx}{u^3}}$ yields

$$G(x,t) = \sqrt{\frac{\Gamma^3}{4\pi\Delta^2 t^3}} \, exp\left(-\frac{\Gamma(t-\Gamma)^2}{4\Delta^2 t}\right) \tag{4}$$

with the two parameters $\Gamma$ and $\Delta$ (Haine and Hall, 2002, both given in years) defining the shape of the TTD. This has been referred to as the inverse Gaussian TTD (IG-TTD), which is essentially a distribution of circulation times or ages. $\Gamma$ is the first moment of the PDF and represents the mean age. $\Delta$ is related to the second moment of the PDF and is a measure of the width of the PDF, i.e., it describes how much a tracer signal disperses during the flow as a result of lateral mixing. The larger this mixing, the more incorporation of waters with different ages occurs within a water parcel sampled downstream from the initialization point of the input function. For $\Delta = 0$, the flow is purely advective. In this case, $\Gamma$ is equal to the tracer age from the binary mixing model at a location $x = u\Gamma$. The ratio $\Delta/\Gamma$ (the Péclet number) is a measure of the advectiveness of the flow (with a smaller $\Delta/\Gamma$ indicating greater advective flow) and typical values range between $0.4 - 1.8$ (e.g., Stöven et al., 2015, for the Fram Strait). $\Delta/\Gamma$ is often set to the value of "one", reflecting an equal contribution from advection and lateral diffusion. This value was found to be a good representation in many parts of the world's oceans and reduces the number of free parameters to one (e.g., Waugh et al., 2004).

In the TTD model, we used the AW input function for the mid-depth Atlantic layer including the entire time range from 1901 up to the sample collection year. In order to determine the two moments of the TTD, $\Delta$ and $\Gamma$ isolines were plotted in the $^{236}U$ vs. $^{129}I$ tracer space (Fig. 2b). More precisely, for a certain $\Gamma$ isoline, $\Gamma$ was fixed and $c(t)$ was calculated for $^{129}I$ and $^{236}U$ concentrations according the Eq. 1 using the $^{129}I$ and $^{236}U$ AW input functions for $c_0(t)$ and varying $\Delta$ such that $0.1 \leq \Delta/\Gamma \leq 1.8$. Plotting the sample data in the same plot allowed for the identification of both $\Delta$ and $\Gamma$ simultaneously, i.e., we did not assume a fixed $\Delta/\Gamma$ ratio. Using the TTD model with the $^{129}I$ - $^{236}U$ tracer pair can provide reliable results for $\Gamma \approx 10 - 70$ years and $\Delta \approx 10 - 50$ years (Fig. 2b). For lower values of both parameters, the isolines intersect due to the shape of the input functions, which are not monotonically increasing or decreasing. For higher values, the isolines converge which also translates into greater uncertainties.

In addition to the mean age $\Gamma$, we also considered the mode age $t_{mode}$. This age is the circulation time with the highest probability within the PDF and has therefore also been referred to as the "most probable" age Smith et al. (2011). It is given by:

$$t_{mode} = \frac{1}{\Gamma}\left(\sqrt{9\Delta^4 + \Gamma^4} - 3\Delta^2\right) \text{ with } t_{mode} \geq 1. \tag{5}$$

For a fixed $\Gamma$, the absolute difference between mode and mean age increases with an increasing $\Delta/\Gamma$ ratio, i.e., with the amount of lateral mixing involved. Note that in our application of the TTD model, we considered tracer input from a single source only and thus neglected the mixing of tracer signals from different sources (which is the principle point of the binary mixing model).

## 3 Results

### 3.1 Pan-arctic distribution of $^{129}$I and $^{236}$U

The compilation of $^{129}$I and $^{236}$U concentrations presented in this study (data in Table S1) allows for a synoptic view of these two tracers across the Arctic Ocean and Fram Strait. Results will be discussed for the surface layer ($10 - 35\,\mathrm{m}$ depth, Fig. 3a and b) and the Atlantic layer (represented by samples from $250 - 300\,\mathrm{m}$ depth in Fig. 3c and d) (see also Table 1). The data points in Fig. 3 represent single measurements, i.e., individual stations and depths.

In the surface layer of the Arctic Ocean, the highest $^{129}$I concentrations ($> 6000 \times 10^6\,\mathrm{at}\cdot\mathrm{l}^{-1}$ were found in the Amundsen Basin in 2015 and the core Fram Strait outflow along the Greenland shelf break in 2016. These locations reflect the pathway for Arctic surface waters originating from the NCC that carries the highest $^{129}$I tracer signal into the Arctic Ocean (Casacuberta et al., 2018). In 2015, high $^{129}$I concentrations were clearly restricted to the Eurasian Basin and the sharp decline across the Lomonosov Ridge towards the Makarov Basin implies the recirculation of AW within the Amundsen Basin. In contrast, $^{129}$I concentrations decreased gradually towards the Makarov and Canada Basin in 2011/12, indicating that AW was probably transported further into the Makarov Basin at that time. Lowest $^{129}$I concentrations in the surface layer (around $100 \times 10^6\,\mathrm{at}\cdot\mathrm{l}^{-1}$) were observed in the Canada Basin, suggesting the minimal presence of Atlantic-origin waters. $^{129}$I concentrations in the Amundsen Basin in 2015/16 samples (Fig. 3a – diamonds) were generally slightly higher compared to those from 2011/12 (Fig. 3a – circles), which might mirror the shape of the $^{129}$I input function which had a sharp increase during the 1990s.

The distribution of $^{236}$U in the surface layer was generally similar to that of $^{129}$I. Highest $^{236}$U concentrations ($> 25 \times 10^6\,\mathrm{at}\cdot\mathrm{l}^{-1}$) were found in the Makarov Basin and the Amundsen Basin in 2011/12, followed by the Fram Strait outflow waters in 2016 (about $20 \times 10^6\,\mathrm{at}\cdot\mathrm{l}^{-1}$). Again, this reflects the presence of AW transported from the NCC carrying the high $^{236}$U tracer signal of the reprocessing plants. The inflowing FSBW and the Nansen Basin showed lower $^{236}$U between $12 - 15 \times 10^6\,\mathrm{at}\cdot\mathrm{l}^{-1}$, which is consistent with the $^{236}$U input function for FSBW in the Atlantic layer (Fig. 1c). As for $^{129}$I, samples taken in the Canada Basin exhibited the lowest $^{236}$U concentrations within the surface layer of $8 \times 10^6\,\mathrm{at}\cdot\mathrm{l}^{-1}$.

At around $300\,\mathrm{m}$ depth in the Atlantic layer, $^{129}$I concentrations in the Eurasian Basin and the Fram Strait were significantly lower compared to the surface layer, with highest concentrations of about $3600 \times 10^6\,\mathrm{at}\cdot\mathrm{l}^{-1}$ observed in the Nansen Basin. In the Canada Basin and Makarov Basin, however, $^{129}$I concentrations were higher compared to the surface layer, around $1000 - 3000 \times 10^6\,\mathrm{at}\cdot\mathrm{l}^{-1}$. A similar pattern was observed for $^{236}$U, whereby the highest $^{236}$U concentration reported in this study was measured in one sample taken at $300\,\mathrm{m}$ depth in the Canada Basin in 2015 (around $45 \times 10^6\,\mathrm{at}\cdot\mathrm{l}^{-1}$). In the rest of the Arctic Ocean, all $^{236}$U concentrations at the same depth level were below $24 \times 10^6\,\mathrm{at}\cdot\mathrm{l}^{-1}$, with highest concentrations detected in the Makarov Basin and the Fram Strait core outflow.

For the Canada Basin, the extremely high $^{236}$U concentrations measured between $250 - 500\,\mathrm{m}$ depth suggest that minimal dilution of the Atlantic layer input occurred during transport. In this regard, it should be noted that the $^{236}$U input function used in this study is largely based on a reconstruction of releases from the reprocessing plants, since no discharge data for $^{236}$U is available for time periods prior to the 1990s (Christl et al., 2015). A recent study used shell archives to reconstruct the time series for early $^{236}$U releases from Sellafield and La Hague (Castrillejo et al., 2020). They found higher $^{236}$U releases (by an

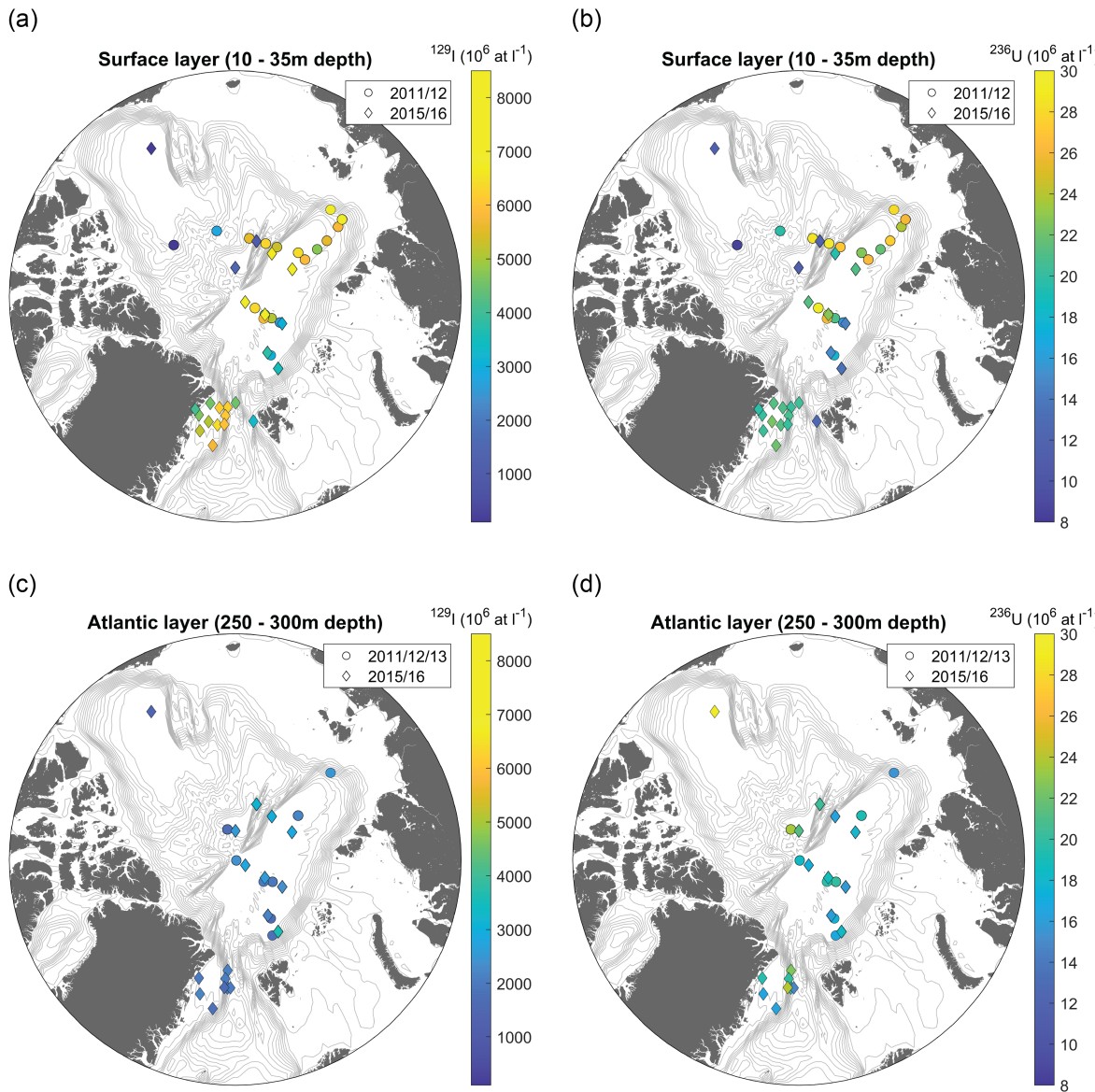

**Figure 3.** $^{129}$I (left) and $^{236}$U (right) concentrations for stations in the Arctic Ocean and Fram Strait used in this study. Results are divided in the surface (a and b) and the Atlantic layer (represented here by $250 - 300\,\text{m}$ depth) (c and d). Different symbols refer to different years, 2011/12/13 in circles and 2015/16 in diamonds. Each data point represents a single measurement (individual station and depth). Note that the same color-scale was applied for surface and Atlantic layer for better comparability between both layers. For the measured $^{129}$I and $^{236}$U concentrations please refer to Table S1 in the supplementary material.

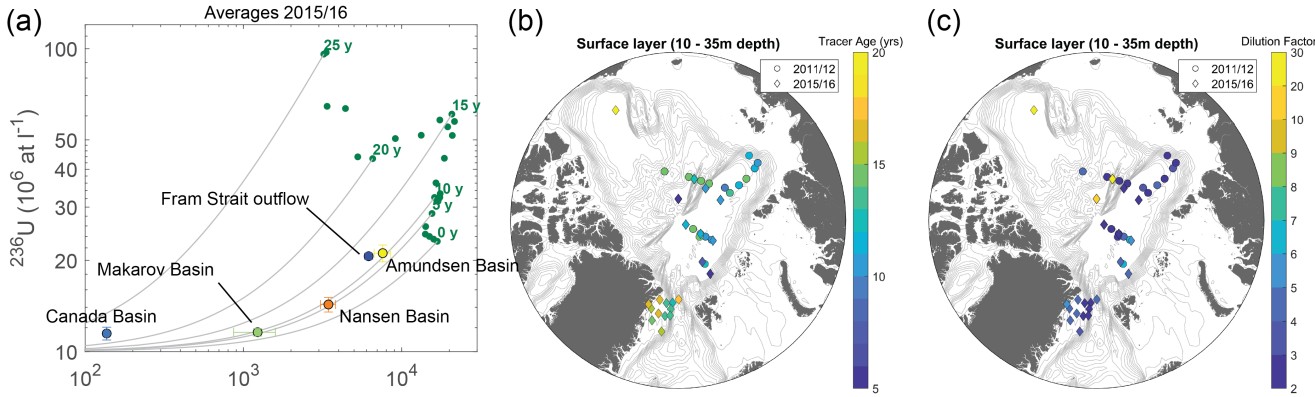

**Figure 4.** (a) Tracer age curves in the binary mixing model ($^{236}$U vs. $^{129}$I) for the surface layer input function (green dots). Tracer ages (based on a collection year of 2015) are depicted in bold numbers. Samples from the surface layer from 2015 and 2016 were sorted by basin (see Table S1) and averaged, the average values and corresponding standard deviations are plotted on top of the mixing curves. Note the log-log scale. (b) Tracer ages and (c) dilution factors determined from the binary mixing model for individual stations in the surface layer.

order of magnitude) from Sellafield during the 1970s compared to those reconstructed by Christl et al. (2015). This finding suggests that the early component of the $^{236}$U input function may have been underestimated, which could explain the high $^{236}$U concentrations measured in the intermediate water samples from the Canada Basin that are presumed to be older than intermediate water samples from other basins.

The differences observed between tracer distributions in the surface and the Atlantic layer reflect the differing pathways of AW between the two layers: In the surface layer, AW is largely restricted to the Eurasian Basin, whereas in the mid-depth Atlantic layer, it is also transported into the Canada Basin. This circulation pattern confirms hydrographic observations (e.g., Rudels, 2015) and highlights the strength of $^{129}$I and $^{236}$U as AW tracers.

### 3.2   Circulation times and dilution factors in the surface layer

The application of the binary mixing model to samples from the surface layer revealed tracer ages on the order of 10-20 years across the Arctic (Fig. 4a). Rather than plotting each individual sample on the binary mixing lines, samples from 2015 and 2016 were here grouped by basins and $^{129}$I and $^{236}$U concentrations were averaged to provide a general overview.

In the central Arctic, tracer ages ranged from 8 years in the Makarov Basin to 12 years in the Fram Strait. The oldest tracer age of 22 years was found in the Canada Basin (Fig. 4a). This station, however, had also the strongest dilution with the global

fallout endmember (DF > 20 or less than 5 % of the input function signal), hence the determination of a tracer age was less precise. The same holds true for the Makarov Basin. Less dilution with global fallout was found in the Amundsen Basin and the Fram Strait (DF of 2-3), which allowed for a more robust determination of tracer ages. A consideration of uncertainties on tracer ages and dilution factors arising from uncertainties in the input function can be found in Appendix A.

To investigate the spatial and temporal pattern of surface water tracer ages in greater detail, ages were constrained for individual stations and for all sampling years (Fig. 4b). Here, samples with a dilution factor larger than 20 were considered to be primarily composed of Pacific origin water and were not included in the subsequent analyses. Taking the results from all expeditions into account provides a better spatial coverage of the Arctic Ocean, including samples from the eastern Amundsen Basin, close to the Laptev Sea shelf.

An overall increase in tracer ages was observed from close to the Laptev Sea shelf across the Amundsen Basin and towards the Fram Strait outflow. This pattern reflects the transport of surface waters originating from the NCC. Dilution factors constrained for the Arctic Ocean and Fram Strait generally ranged between 2 and 6 (Fig. 4c). The lowest dilution of tracer-labeled Atlantic origin surface waters with the global fallout background was observed for samples closest to the Laptev Sea, in the Amundsen Basin and in the core Fram Strait outflow (DF between 2 and 3). This pattern is consistent with advective transport of NCC origin water across the Arctic Ocean. In the Nansen Basin, dilution factors were slightly higher (up to 6.5). This might be due to mixing of the NCC signal with mainly global fallout labeled AW from the NwAC transported into the Nansen Basin through Fram Strait, as for instance suggested in Karcher et al. (2012).

Regarding differences between sampling years (2011/12 vs. 2015), a slight decrease of tracer ages over time could be observed for the central Arctic Ocean (Fig. 4b). In the Makarov Basin, dilution factors significantly increased from about 2 in2011/12 to 10-20 in 2015. This implies that the Pacific Water fraction increased over time and can be explained by a shift of the Atlantic-Pacific water front that generally determines the water mass provenance in the Makarov Basin. The alignment of the Atlantic-Pacific front across the Arctic Ocean is influenced by the prevailing atmospheric circulation (indicated by, e.g., the Arctic Oscillation (AO) index; Morison et al. (2012); Alkire et al. (2019) and ref. therein). However, no clear trend in the AO index was observed between 2011/12 and 2015 and hence the reason behind changes in the Pacific Water fraction in the Makarov Basin remains unclear. Tracer ages should not be affected by the water mass provenance but could change as a result of potentially changing AW pathways in the central Arctic Ocean.

In the Fram Strait, tracer ages of shelf break stations in the core of the EGC were 12-13 years, whereas towards the coast, tracer ages increased up to 17 years (Fig. 4b). A similar pattern was observed for dilution factors, with less dilution in the stations along the shelf-break (coinciding with lower tracer ages) and higher dilution factors closer to the coast of Greenland. This indicates that a recirculation of waters could have occurred on the shelf, as previously proposed in Wefing et al. (2019). Additionally, melt waters from Greenland glaciers may have influenced the results from the coastal stations. Closer towards the coast, samples did not seem to dilute with the global fallout endmember but rather seemed to be influenced by a different endmember with a higher $^{236}$U concentration (see also Fig. 5 in Wefing et al. (2019)), which led to increased tracer ages in the binary mixing model. Due to a lack of $^{129}$I and $^{236}$U data from Greenland glacier melt water, however, we could not investigate this hypothesis further.

### 3.3 Circulation times and mixing regimes in the mid-depth Atlantic layer

The mean ages ($\Gamma$) inferred from the TTD method for the mid-depth Atlantic layer were overall larger than the tracer ages inferred from the binary mixing model for the surface layer (Fig. 5a). For an overview about the TTD parameters, stations

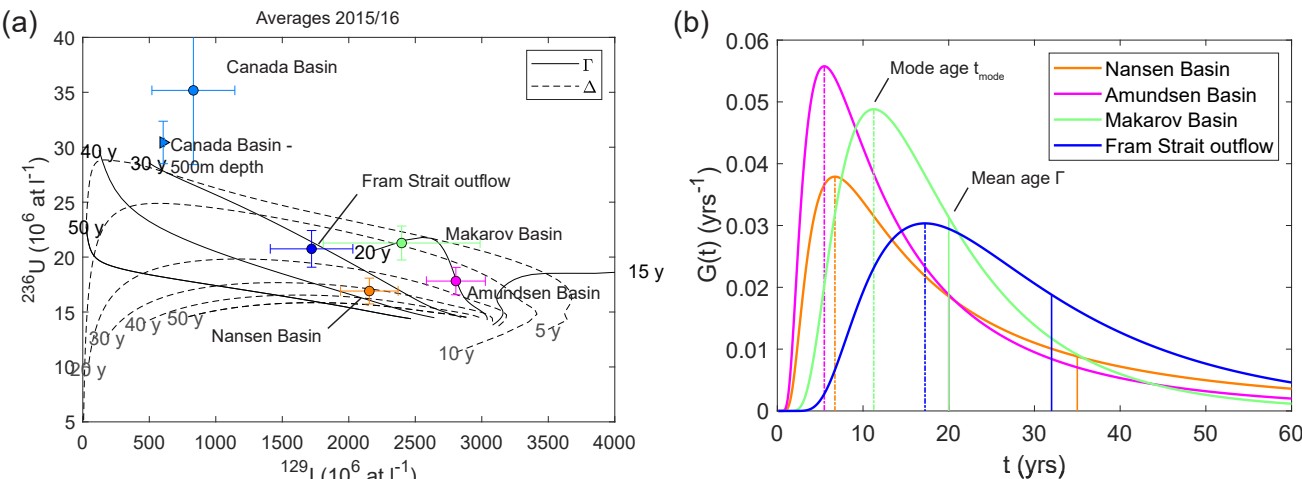

**Figure 5.** (a) $\Gamma$ (solid) and $\Delta$ (dashed) isolines in the TTD model ($^{236}$U vs. $^{129}$I) calculated assuming an inverse Gaussian TTD (sampling year 2015) and the Atlantic layer input function. Samples from 2015 and 2016 ($250-500\,\mathrm{m}$ depth) were sorted by basin (see Table S1) and averaged, the average values and corresponding standard deviations are plotted on top of the isolines. (b) Inverse Gaussian TTDs for the different Arctic basins based on the $\Delta$ and $\Gamma$ values determined from a). Dotted vertical lines indicate the mode age, solid lines the mean age for each basin.

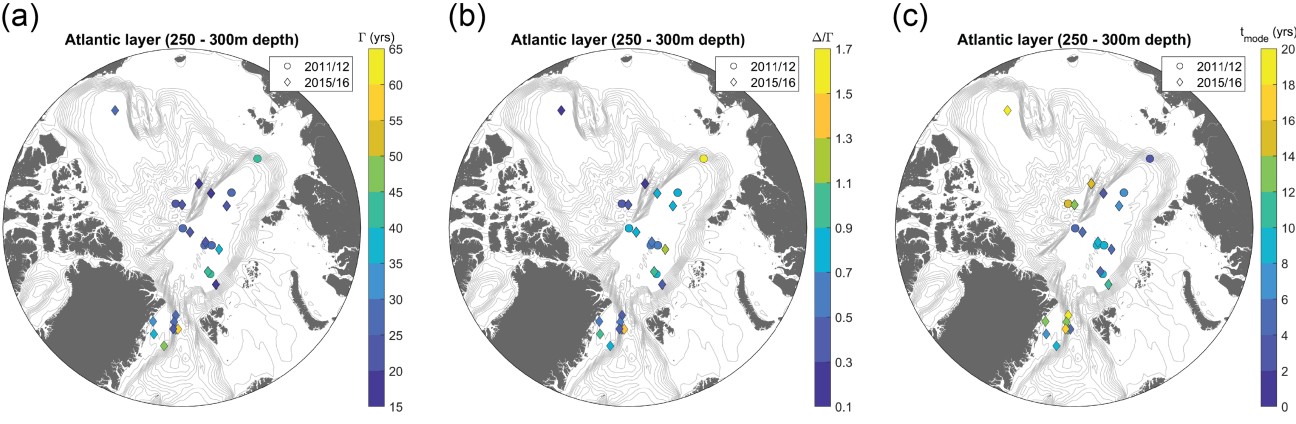

**Figure 6.** a) Mean ages $\Gamma$, b) $\Delta/\Gamma$ ratios and c) mode ages $t_{mode}$ determined from the TTD model for individual stations in the Atlantic layer ($250-300\,\mathrm{m}$ depth and $500\,\mathrm{m}$ depth for the Canada Basin).

were first grouped by basins as in 3.2. Here, all samples from the mid-depth Atlantic layer ($250-800\,\mathrm{m}$ depth) were averaged. Uncertainties on $\Gamma$ and $\Delta$ are discussed in Appendix A.

Mean ages were lowest (about 20 years) in the Makarov and Amundsen Basins and higher (about 35 years) in the Nansen Basin and the Fram Strait outflow. The values of $\Delta$ tend to correlate positively with the mean age, with the lowest $\Delta$ of 9 years

determined for the Makarov Basin, followed by the Amundsen Basin and the Fram Strait (15 years) and the Nansen Basin (32 years). An average $\Gamma$ and $\Delta$ could not be constrained for the Canada Basin, as the average value from $250 - 800\,\mathrm{m}$ depth plots outside the TTD model domain (Fig. 5a). Therefore, the datapoint from $500\,\mathrm{m}$ depth was plotted instead, showing a mean age of about 30 years and a $\Delta$ of about 5 years at that depth. The inferred ratio of $\Delta/\Gamma$ varied substantially across basins, but always stayed below 1, i.e., was indicative of a rather advective flow. The lowest value (0.5) of $\Delta/\Gamma$ was found in the Makarov Basin and the Fram Strait outflow, followed by the Amundsen Basin (0.75). The highest ratio ($\Delta/\Gamma = 0.9$) was determined for the Nansen Basin. Mode ages ($t_{mode}$) derived from $\Gamma$ and $\Delta$ were lowest (5-7 years) in the Nansen and Amundsen Basins and increased towards the Makarov Basin (11 years) and the Fram Strait (17 years).

From the $\Delta$ and $\Gamma$ values determined for each basin the corresponding IG-TTDs were constructed according to Eq. 4 (Fig. 5b). The shape of these distributions and especially the span between mean and mode ages clearly illustrate the different flow regimes in the different Arctic basins: In the Nansen Basin, the distribution was highly asymmetric and the difference between mean and mode age was largest (almost 30 years). The long tail towards older ages indicated the presence of very old water parcels. In contrast, the Makarov Basin TTD was more symmetric and the two age measures differed by less than 10 years. This suggests that less mixing was involved in the transport of AW to the Makarov Basin.

Overall, the mean age distribution (Fig. 6a; here depicted at $250 - 300\,\mathrm{m}$ depth) of individual stations showed a similar pattern as the tracer ages in the surface layer. Ages increased along the expected AW flow in the mid-depth Atlantic layer (e.g., Rudels, 2015), with younger waters in the Eurasian Basin and older waters in the Canada Basin and the Fram Strait outflow. A similar pattern was also observed in the distribution of mode ages (Fig. 6c). Across the Lomonosov Ridge, mean ages were comparable but significant differences were evident in the mode ages as well as the $\Delta/\Gamma$ ratios (Fig. 6b), showing greater advective flow in the Makarov Basin compared to the Amundsen Basin.

A high mean age and the highest $\Delta/\Gamma$ ratio were found for one station close to the Laptev Sea shelf. This result was rather unexpected given the proximity of this station to the initialization point of the input functions and was due to the comparably low $^{129}$I and $^{236}$U concentrations measured at $300\,\mathrm{m}$ depth. A possible explanation could be a higher influence of the FSBW input function at that depth, rather than the mixture of FSBW and BSBW that is used for our Atlantic layer input function. Using the FSBW input function instead, the mean age for this station would be on the order of 10 years and the $\Delta/\Gamma$ ratio would be about 0.5, which is much more reasonable. In the context of this study, the Laptev Sea shelf data point will therefore not be discussed further.

In the Fram Strait, a similar pattern was observed in the mid-depth Atlantic layer as for the surface layer: Lower mean ages and $\Delta/\Gamma$ ratios around 0.5 along the shelfbreak indicate greater advective flow in the EGC core whereas $\Gamma$ and $\Delta/\Gamma$ increased towards the coast of Greenland.

Considering individual samples from the entire Atlantic layer (from $250 - 800\,\mathrm{m}$ depth) on a transect across the Arctic Ocean (Fig. S2a and b), we observed that the high mean ages and $\Delta/\Gamma$ ratios in the Nansen Basin were not limited to $300\,\mathrm{m}$ depth, but were also obtained for samples from $500\,\mathrm{m}$ depth. Whereas mean ages and $\Delta/\Gamma$ ratios in the Amundsen and Makarov Basins did not change with depth, an increase of mean ages up to more than 50 years at $800\,\mathrm{m}$ depth was observed in the Fram Strait. In the Canada Basin, mean ages and $\Delta/\Gamma$ ratios could be determined for samples from 500 and $800\,\mathrm{m}$ depth and lay around

30 years and 0.1, respectively. This implies that across the Atlantic layer of the Arctic Ocean, $\Delta/\Gamma$ ratios were lowest in the Canada Basin, suggesting very advective AW flow. Mode ages were more or less constant over depth, a slight increase was only observed in the Makarov Basin (Fig. S2c). Largest mode ages of around 30 years were obtained for the Canada Basin at 500 and 800 m depth.

## 4  Discussion

In the first part of the discussion, the use of $^{236}$U (in combination with $^{129}$I) will be validated as a transient tracer to estimate circulation times in the Arctic Ocean. This will be done in the context of earlier studies that used different models and tracers. The results presented in this paper will be mainly compared to the study by Smith et al. (2011), which is also based on the use of two anthropogenic radionuclides ($^{129}$I and $^{137}$Cs). The TTD model results determined in this work will additionally be compared to other studies conducted in the Arctic Ocean that used CFCs and SF$_6$ as ventilation tracers. A second subsection of the discussion examines the use of the mode age as a measure of circulation times and conceptual differences between mean and mode ages will be discussed. The last part of the discussion highlights the implications of the obtained circulation times for a better understanding of AW circulation in the Arctic Ocean.

### 4.1  Circulation patterns of Atlantic Water in the Arctic Ocean revealed by $^{129}$I and $^{236}$U

#### 4.1.1  Circulation times of Atlantic Water in the context of earlier studies

For the central Arctic Ocean, tracer ages and mean ages obtained from the combination of $^{129}$I and $^{236}$U in the mixing model and the TTD model, respectively, are overall in good agreement with those presented in Smith et al. (2011) (based on samples collected in the mid-1990s), especially in the Amundsen Basin (Table 2). For both, the surface layer and the mid-depth Atlantic layer, AW ages generally increased from the entrance of the Arctic Ocean through the Amundsen and Makarov Basins towards the Fram Strait. This pattern is consistent with the general understanding of AW flow in the Arctic Ocean derived from hydrographic measurements (e.g., Rudels, 2015). $^{129}$I-$^{236}$U mean ages for the Fram Strait are in line with mean ages observed slightly upstream (north of Greenland) by Smith et al. (2011). In the Fram Strait, the observed increase of mean ages from 300 to 800 m depth indicates that the mean transport was slower in the BSBW layer compared to the FSBW layer. This result, together with the fact that increased mean ages were not observed in the BSBW layer of the Amundsen Basin, supports the common perception that BSBW tends to follow the path of the Arctic Ocean Boundary Current through the Canada Basin whereas FSBW rather returns to the Fram Strait within the Eurasian Basin, along the flanks of the Lomonosov Ridge (e.g., Rudels, 2015).

#### 4.1.2  Dilution factors and the Atlantic-Pacific Water interface in the surface layer of the Makarov Basin

Dilution factors obtained from the binary mixing model for the surface layer reflect mixing between tracer-labeled AW and waters carrying the global fallout background. The results determined in this study agree with those reported in Smith et al.

**Table 2.** Tracer ages (TA), dilution factors (DF), mean ages ($\Gamma$) and $\Delta/\Gamma$ ratios determined from the two age models for the different basins of the Arctic Ocean in Polar Surface Water (PSW) and the mid-depth Atlantic layer are compared to those from earlier studies. Given ranges correspond to minimum and maximum values for stations within the basin for all sampling years. For comparison to Smith et al. (2011) tracer ages and dilution factors were normalized to the entrance of the Arctic Ocean (dilution factors have been divided by 3 to account for dilution from 60°N, where the input function from Smith et al. (2011) was defined, to the Barents Sea). For Tanhua et al. (2009) and Stöven et al. (2016), the time t=0 is set by the isolation of waters from the atmosphere, which was here approximated as the Barents Sea opening. AAW: Arctic Atlantic Water, Init. point: Initialization point.

| Depth/Water mass | PSW | | PSW | | 250-300 m | | 200-300 m | | 425 m | | AAW | |
|---|---|---|---|---|---|---|---|---|---|---|---|---|
| Reference | This study | | Smith et al. (2011) | | This study | | Smith et al. (2011) | | Tanhua et al. (2009) | | Stoven et al. (2016) | |
| Tracers | $^{129}I$, $^{236}U$ | | $^{129}I$, $^{137}Cs$ $^{129}I$, CFC-11 | | $^{129}I$, $^{236}U$ | | $^{129}I$, $^{137}Cs$ $^{129}I$, CFC-11 | | CFCs, SF$_6$ | | CFC-12, SF$_6$ | |
| Model | Mixing model | | Mixing model | | TTD model | | TTD model | | TTD model | | TTD model | |
| Init. point (t=0) | 74°N | | Barents Sea opening | | 74°N | | Barents Sea opening | | Barents Sea opening | | Barents Sea opening | |
| | TA | DF | TA | DF/3 | $\Gamma$ | $\Delta/\Gamma$ | $\Gamma$ | $\Delta/\Gamma$ | $\Gamma$ | $\Delta/\Gamma$ | $\Gamma$ | $\Delta/\Gamma$ |
| | (yrs) | | (yrs) | | (yrs) | | (yrs) | | (yrs) | | (yrs) | |
| Nansen Basin | 3-12 | 4-6.5 | 4-9 | 2.5-4 | 15-45 | 0.3-1.2 | 10-30 | 1.0-2.0 | >35 | 1 | - | - |
| Amundsen Basin | 9-16 | 1.5-3 | 7-9 | 1.5-2 | 18-28 | 0.6-0.9 | 20-30 | 0.5-1.0 | 15-35 | 1 | - | - |
| - North Pole | 11 | 2.5 | 9-12 | 2-3 | 21 | 0.8 | - | - | - | - | - | - |
| - near Laptev Sea | - | - | - | - | 41* | 1.6* | 13 | 2.0 | 5-15 | 1 | - | - |
| Makarov Basin | 2-14 | 2-20 | 4-12 | 1.5-4 | 18-23 | 0.3-0.4 | 18-22 | 0.4-0.6 | 30-50 | 1 | - | - |
| Canada Basin | 14-20 | 4-20 | - | - | 29** | 0.1** | 20-30 | 0.3-0.4 | 50-60 | 1 | - | - |
| North of Greenland | - | - | - | - | - | - | 33-36 | 0.2-0.3 | 50-55 | 1 | - | - |
| Fram Strait | 12-17 | 3-5 | - | - | 24-55 | 0.3-1.4 | - | - | - | - | 32 ± 51 | 1 |

* single data point

** values from 500 m depth

(2011) (Table 2). Note that dilution factors calculated by Smith et al. (2011) were referenced to 60°N, whereas in the present study, the input function was defined at the entrance to the Arctic Ocean. Hence, dilution factors from Smith et al. (2011) were corrected by the dilution factor of 3, that was determined for one station in the Barents Sea opening.

Low dilution factors in the Amundsen Basin and the core Fram Strait outflow suggest a confined transport of tracer-labeled waters originating from the NCC into the central Arctic Ocean and out through the Fram Strait. The higher dilution factors in the Nansen Basin indicate the presence of waters labelled only by the global fallout signal, either old AW or waters originating from the NwAC (note that dilution with Pacific Water is unlikely to happen in the Nansen Basin).

     In the Makarov Basin, the global fallout signal is mainly transported by water of Pacific-origin, hence the dilution factors

can be interpreted as a measure of the Pacific Water fraction. From 2011/12 to 2015, a significant increase in dilution factors from 2 to 20 was observed in the Makarov Basin, hence indicating an increase in the Pacific Water fraction. Note that further downstream, a change in the spreading of Pacific Water could also be reflected in the composition of outflowing waters in the Fram Strait, but no time-series of $^{129}$I and $^{236}$U data is available for the Fram Strait to date. The finding of a high Pacific Water fraction in the surface of the Makarov Basin in 2015 is supported by estimates from nutrient relationships (N:P ratios) from

the same year (Alkire et al., 2019, "West leg" of the GEOTRACES GN01 expedition), that determined Pacific Water fractions greater than 70%. New tracers that can be used to differentiate between Atlantic- and Pacific derived waters are a valuable tool because N:P ratios, used as the classical Atlantic-Pacific Water tracer, are known to be influenced by processes occurring along the transport, especially denitrification in the broad shelf seas of the Arctic Ocean (Bauch et al., 2011; Alkire et al., 2019). In contrast, $^{129}$I and $^{236}$U behave more conservatively than nutrients and are not affected by any biogeochemical processes taking

place during transport, suggesting the potential to use these two anthropogenic radionuclides as water mass provenance tracers in the surface Arctic Ocean.

### 4.1.3    Lateral mixing within the Atlantic Water flow and its impact on mean ages

For the mid-depth Atlantic layer, along-flow lateral mixing taking place within the AW flow is described by the $\Delta/\Gamma$ ratio and influences the mean ages obtained from the TTD model. $\Delta/\Gamma$ ratios determined from $^{129}$I and $^{236}$U agree well with the

findings from Smith et al. (2011) for all basins (Table 2). Overall, the AW flow was found to be rather advective, especially in the Makarov and Canada Basins and the core of the Fram Strait outflow.

     An exception is the Nansen Basin, where $\Delta/\Gamma$ ratios $\geq 1$ were determined for two stations (Fig. 6b), which coincided with high mean ages (Fig. 6a). Especially the high mean ages in the Nansen Basin were a rather unexpected finding, as the stations are located close to the tracer sources in the Fram Strait and the Barents Sea (i.e., one would expect younger waters that have

not undergone much lateral mixing). High $\Delta/\Gamma$ ratios imply a broad TTD and significant mixing of waters with different ages. One possible explanation for the rather high $\Delta/\Gamma$ ratios found in the Nansen Basin resulting in high mean ages is a mixture of inflowing AW in the FSBW with water that has been transported through the Arctic and re-entered the Nansen Basin as a result of recirculation north of Fram Strait. In a similar process, the direct recirculation of northward flowing FSBW within Fram Strait might have led to the elevated $\Delta/\Gamma$ ratios observed in the eastern- and southernmost stations of the Fram Strait

which are also accompanied by higher mean ages.

Two recent ventilation studies in the Arctic Ocean using the TTD model with atmospherically introduced transient tracers assumed a fixed $\Delta/\Gamma = 1$ (Tanhua et al., 2009; Stöven et al., 2016). Mean ages for the Nansen and Amundsen Basin and the Fram Strait determined from these studies are in general agreement with the $^{129}$I - $^{236}$U TTD model results (Table 2). Across the Lomonosov Ridge, however, Tanhua et al. (2009) observed a sharp front for all samples below $200\,$m depth with an increase of mean ages in the direction of the Makarov Basin, where mean ages were significantly higher compared to the present $^{129}$I - $^{236}$U study. In this regard it should be noted, that a recent model study suggested a general overestimation of CFC-derived mean ages, especially in waters ventilated in the Barents Sea, due to an undersaturation of pCFC-12 in the surface waters (Terhaar et al., 2020b).

In the Makarov Basin, also the greatest discrepancy between the uniform $\Delta/\Gamma = 1$ assumed by Tanhua et al. (2009), and the present data ($\Delta/\Gamma$ of 0.2-0.4) was observed. Consequently, the assumption of a uniform $\Delta/\Gamma = 1$, that was found to be representative of many parts of the world's oceans, does not appear valid for the entire Arctic Ocean, especially for the Makarov Basin. Here the AW flow was found to be rather advective with less lateral mixing occurring compared to the Eurasian Basin. This result is also in line with the recent study by Rajasakaren et al. (2019), who constrained a $\Delta/\Gamma$ of about 0.6 for the Amerasian Basin, based on the comparison between mean ages derived from CFC-12 and SF$_6$. To provide the best estimate of mean ages, a combination of different tracers is required to constrain $\Delta/\Gamma$ separately, which supports the findings from Smith et al. (2011).

Generally, some limitations should be considered with respect to the use of the TTD model in the Arctic Ocean and the Fram Strait in particular. In the Fram Strait, highly advective flow (low $\Delta/\Gamma$ ratios, Fig. 6b) was observed for the EGC core in addition to slightly higher mean ages compared to those in the central Arctic (Fig. 6a). However, an important consideration for the Fram Strait is that it likely registers a mixture of Atlantic-origin water masses composed of those that circulated through the short Eurasian Basin loop and those that followed a pathway through the Canada Basin, i.e., the long loop. As Smith et al. (2011) noted, the IG-TTD is a unimodal distribution which assumes a single dominant pathway in the water transport. This does not hold for the Fram Strait and the results from the simple TTD approach applied in this study should therefore be regarded with some reservation. In a successive approach, a bimodal TTD which would capture both circulation loops could be considered for the Fram Strait (see e.g., Stöven and Tanhua, 2014). The functional form of the Green's function $G(t)$, describing the propagation of the tracer signal, introduces an uncertainty on the reported mean and mode ages. Examples of different functional forms have for instance been discussed in Haine and Hall (2002). The inverse Gaussian form of $G(t)$, which solves the 1D advection-diffusion equation, has been shown to perform well for many oceanographic applications, and is therefore the most widely used functional form.

**Table 3.** Mode ages ($t_{mode}$) determined from the TTD model for different basins of the Arctic Ocean in the Atlantic layer (250-300m depth) in comparison to circulation times from the propagation of hydrographic properties in the FSBW (data and modelling studies). Given ranges correspond to minimum and maximum values of stations within the basin, taking all sampling years into account. WSC: West Spitsbergen Current, T: Temperature.

| Depth/Water mass | 250-300 m | BSBW | FSBW | FSBW | FSBW | FSBW |
|---|---|---|---|---|---|---|
| Reference | This study | Mauldin et al. (2010) | Swift et al. (1997) | Rudels et al. (2000) | Dmitrenko et al. (2008) | Polyakov et al. (2011) |
| Tracers | $^{129}$I, $^{236}$U | $^3$H, $^3$He | T anomalies | T anomalies | T anomalies | T anomalies |
| Model | TTD model | Leaky pipe model | - | - | - | - |
| Init. point (t=0) | 74°N | St. Anna Trough | WSC | WSC | WSC | WSC |
| | $t_{mode}$ (yrs) | $t_{adv}$ (yrs) | time (yrs) | time (yrs) | time (yrs) | time (yrs) |
| Nansen Basin | 4-11 | - | - | - | - | - |
| Amundsen Basin | 4-10 | 3-4 | 10 (North Pole) | 6 | 6 | < 10 |
|   - near Laptev Sea | 3* | 1-2 | - | 5 | 5 | 5 |
| Makarov Basin | 12-15 | - | - | - | - | - |
| Canada Basin | 29** | 6-8 | - | - | - | 10-15 |
| North of Greenland | - | - | - | 7 | - | - |
| Fram Strait | 5-18 | - | - | 10 | - | - |

\* single data point

\*\* values from 500 m depth

## 4.2 Different age measures determined from the TTD model

### 4.2.1 Mode ages in the context of Arctic Ocean circulation

To our knowledge, mode ages of the TTD have only been considered in Smith et al. (2011), who suggested that the mode age might be a better comparison to the tracer age for pulse-like input functions. They only constrained the mode age for a single station, however, where they found it was similar to the tracer ages obtained from the binary mixing model.

A somewhat similar approach is the study by Mauldin et al. (2010) who used the combination of $^3$H and $^3$He to estimate the boundary current transport of BSBW. They applied a leaky pipe model with an advective core and considered mixing with the exterior region (based on Waugh and Hall (2005)) instead of the classical TTD model. The assumed distribution does not have the shape of an inverse Gaussian but peaks at the "advective time" $t_{adv}$ which is calculated from the modeled current velocity of the core region. The overall shape of the distribution with a peak at low transit times and a long tail towards higher transit times is however similar to the IG-TTD and as a first approximation, this advective time can be compared to the mode ages derived in the present study. The advective times reported by Mauldin et al. (2010) for the BSBW pathway along the marginal

slopes of the Amundsen Basin, Makarov, and Canada Basins generally agree with the mode ages obtained in the present study (Table 3). For the Canada Basin, the $^{129}$I - $^{236}$U mode ages are about 25 year higher (i.e., implying longer circulation times) compared to the Mauldin et al. (2010) study.

Here it should be noted that the latter employed data from the 1990s when the Arctic Oscillation was in a positive phase and exceptionally strong cyclonic boundary current conditions prevailed in the Arctic Ocean, associated with accelerated boundary current flow (Karcher et al., 2012). This implies low advective ages. The Canada Basin mode age obtained from $^{129}$I and $^{236}$U for 2015 cannot unambiguously be attributed to the boundary current. Additionally, AW pathways to the Canada Basin probably changed as a consequence of a weakening of the boundary current due to the transition of the AO to a more negative

phase (Karcher et al., 2012).

Mode age results for the Arctic Ocean can also be compared to estimates of circulation times based on the propagation of hydrographic properties. Several studies have for example used temperature anomalies as tracers for AW flow (Swift et al., 1997; Rudels et al., 2000; Dmitrenko et al., 2008; Polyakov et al., 2011). Warm pulses were observed in inflowing AW on hydrographic sections or at moorings in the Fram Strait and subsequently tracked through the Arctic Ocean and re-analyzed

at downstream sampling sites. Circulation times deduced from several studies are summarized in Table 3. In addition, model experiments (e.g., Karcher et al., 2011) suggest that warm water pulses entering the Arctic Ocean through Fram Strait take about 15 years to recirculate back to Fram Strait via the Eurasian Basin loop and about 20-30 years via the Canada Basin loop. These circulation times, both from observations and models, are overall within the same range as mode ages obtained from the combination of $^{129}$I and $^{236}$U.

**4.2.2   Comparison of mean and mode ages**

Mean and mode ages provide different circulation timescales and also impart a different understanding of the flow of AW in the mid-depth Atlantic layer. The mean age of the TTD is mainly influenced by the long tail of the distribution that extends towards old ages. The asymmetry depends on the $\Delta/\Gamma$ ratio as a higher influence of mixing leads to a stronger dispersion of the TTD peak. Mean ages are classically used as the age measure derived from the TTD, especially in ocean ventilation studies

that often aim at an estimation of the uptake of anthropogenic Carbon ($C_{ant}$) from the atmosphere. In this regard, the focus lies on the $C_{ant}$ inventory accumulating over time rather than the determination of tracer signal transport times. In contrast, mode ages reflecting the transit time with the highest probability may better reflect the propagation of biogeochemical pulses (e.g., temperature-salinity anomalies, plankton blooms, etc.).

In general, there is no unique definition of a circulation time once the effect of mixing is taken into account. With respect

to water mass ages, tracer ages derived from the simple mixing model are a valid approximation if the mixing of ages within a water parcel can be assumed to be limited to a short time range, that is if the $\Delta/\Gamma$ ratio is low. In this case, the different age measures deduced from the TTD are also similar and can be compared to tracer ages. If significant mixing is involved, only the TTD model provides meaningful age estimates. We found that overall, mode ages derived from the TTD model for the Atlantic layer are in the same range as tracer ages from the surface layer.

## 4.3 Future perspectives on the use of $^{129}$I and $^{236}$U in the Arctic Ocean

Climate change will continue triggering large scale circulation changes in the Arctic Ocean, fostering the use of transient tracers which can unambiguously label AW and can provide information about circulation pathways and timescales. The present study highlights the combination of $^{129}$I and $^{236}$U in the Arctic Ocean as a powerful tool to understand such processes.

In addition to circulation times and mixing regimes obtained from tracer measurements, simulations using regional ocean models can predict changes in AW circulation patterns in the coming years. A numerical model used to simulate $^{129}$I transport through the Arctic Ocean, for example, revealed significant circulation changes during the mid-1990s and 2000s resulting from changing Arctic Oscillation phases (Karcher et al., 2012). The inclusion of different tracers (e.g., $^{236}$U) into these models, together with a better data coverage, will help to improve our understanding of the Arctic Ocean circulation and ventilation as well as potential responses to climate change.

The properties of AW flow also play a key role in understanding and predicting the transport of heat or $C_{ant}$ to the Arctic Ocean. A recent study suggests that the amount of $C_{ant}$ that entered the Arctic Ocean through lateral transport, i.e., inflowing AW, is about three times larger compared to what entered through air-sea gas exchange (Terhaar et al., 2019). Latest estimates predict a $C_{ant}$ inventory of about $9\,\mathrm{Pg}$ Carbon at the end of the 21st century, which exceeds projections from earlier studies (Terhaar et al., 2020a). The consequent decrease in calcium carbonate saturation states is suggested to occur in the mid-depth Atlantic layer of the Arctic Ocean rather than the surface waters, in depths between $400-800\,\mathrm{m}$, according to Terhaar et al. (2020a). A study by Ulfsbo et al. (2018) found a strong linear correlation between measurements of $C_{ant}$ and $^{129}$I in the Atlantic layer of the Eurasian Basin in 2011/12 and 2015, also pointing to AW as the main source of both substances. Circulation times obtained from the combination of $^{129}$I and $^{236}$U can therefore help understanding the fate of $C_{ant}$ and predict its propagation into the different Arctic basins.

## 5 Conclusions

$^{129}$I and $^{236}$U measurements on samples collected in the Arctic Ocean and Fram Strait between 2011 and 2016 were used to investigate circulation timescales and flow characteristics of AW in the Arctic Ocean. Two different models were utilized: a binary mixing model for the surface layer and the transit time distribution model for Atlantic Water in the mid-depth Atlantic layer.

The binary mixing model, which takes into account the mixing between AW labeled with a radionuclide signal from European nuclear reprocessing plants and waters carrying background concentrations from global fallout, provided AW tracer ages which support findings from previous studies (summarized in Fig. 7, green). This is a useful finding as it is problematic to use atmospherically introduced ventilation tracers to study circulation timescales of surface waters which are constantly exposed to atmospheric gas exchanges. The $^{129}$I and $^{236}$U tracer pair has been shown to be conservative with a well defined, geographically specific input function that incidentally provides a method for the accurate resolution of Atlantic-origin from Pacific-origin water in the surface layer of the Makarov Basin. For this location, dilution factors obtained from the mixing model suggested an increase in the Pacific Water fraction between 2011/12 and 2015.

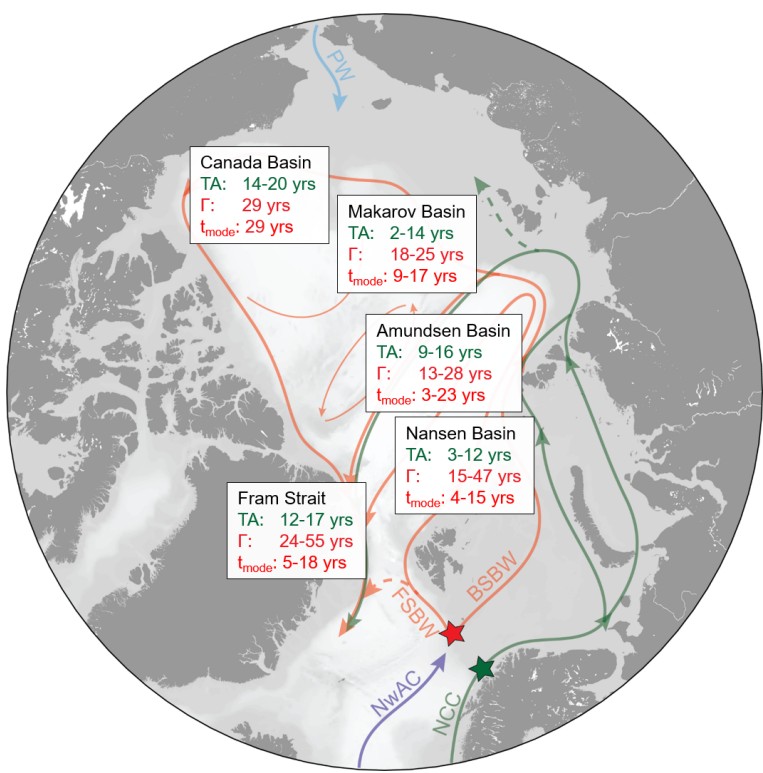

**Figure 7.** Summary of circulation times of Atlantic Water in the Arctic Ocean from this study: Tracer ages (TA) derived from the binary mixing model for the surface layer (green), mean ages ($\Gamma$) and mode ages ($t_{mode}$) derived from the TTD model for the mid-depth Atlantic layer (red).

In the mid-depth Atlantic layer ($250 - 800\,\text{m}$ depth), the TTD model was used to evaluate the flow of AW through the major Arctic basins. Mean ages ($\Gamma$, Fig. 7, red) inferred from the TTD model were in good agreement with those obtained from ventilation tracer studies for Arctic Ocean basins where $\Delta/\Gamma$ ratios from $^{129}$I and $^{236}$U matched the assumed $\Delta/\Gamma = 1$ from other studies. This assumption did not appear to be valid for the entire Arctic Ocean, however, especially in the Makarov and Canada Basins, where $\Delta/\Gamma$ ratios were significantly smaller than one.

Mode ages, which correspond to the most probable age of the TTD, were in the same range as circulation times deduced from the propagation of temperature anomalies ($t_{mode}$, Fig. 7, red). For the lateral propagation of tracer signals, especially those of a pulsed nature, the mode age may be a better age measure compared to the mean age, as it is not influenced by the asymmetric shape of the TTD and the long tail towards old ages. For a comparison between tracer ages, mean, and mode ages, the different underlying models for the surface and the Atlantic layers need to be considered. In the case of lateral mixing, tracer ages are not meaningful and mean ages will always exceed mode ages. Here, the choice of the age estimate generally depends on the context.

Overall, AW pathways and circulation times constrained from the combination of $^{129}$I and $^{236}$U add to a better understanding of the present state of Arctic Ocean circulation. The Arctic is considered a region to experience major changes as a consequence of global warming in future years, including higher freshwater input due to sea-ice melt, increased heat transported into the Arctic Ocean by AW or ocean acidification due to the uptake of anthropogenic carbon. Hence transient tracers labeling AW are valuable tools to track the future evolution of AW circulation in the Arctic Ocean.

*Data availability.*  Data used in this study can be found in Table S1 in the supplementary material.

## Appendix A: Uncertainties

Uncertainties for basin-averaged tracer ages and dilution factors as well as the TTD parameters result from both the standard deviation of the averaged samples (error bars in Fig. 4a and 5) and from uncertainties in the $^{129}$I and $^{236}$U input functions (Fig. 1b and c, about 10-20 % for the last 30 years). Due to the irregular shape of the input functions the resulting uncertainties are

highly asymmetric.

To estimate the influence of input function uncertainties, a sensitivity analysis was performed as an example for the Amundsen Basin average from 2015. In the binary mixing model (surface layer, tracer age of 10 years and dilution factor of 2.2), the uncertainty in the $^{236}$U input function was the main driver for tracer age uncertainties (Fig. A1a), whereas dilution factors were mainly affected by uncertainties in the $^{129}$I input function (Fig. A1b). In 2005 (corresponding to tracer age of 10 years

for samples from 2015), the $^{129}$I and $^{236}$U surface layer input functions had associated uncertainties of about 20 % and 10 %, respectively. This translated to approx. $+2/-5$ years uncertainty in the tracer age and $+0.8/-0.6$ in the dilution factor.

In the TTD model (mid-depth Atlantic layer, $\Gamma = 20$ years, $\Delta = 15$ years), the uncertainty in the $^{129}$I input function was the main driver for uncertainties in $\Gamma$ (Fig. A1c) and the uncertainty in the $^{236}$U input function was the main driver for uncertainties in $\Delta$ (Fig. A1d). The actual Atlantic layer input function uncertainties of around 10 % for $^{129}$I and 5 % for $^{236}$U resulted in

uncertainties of the order of $\pm 5$ years for both $\Gamma$ and $\Delta$.

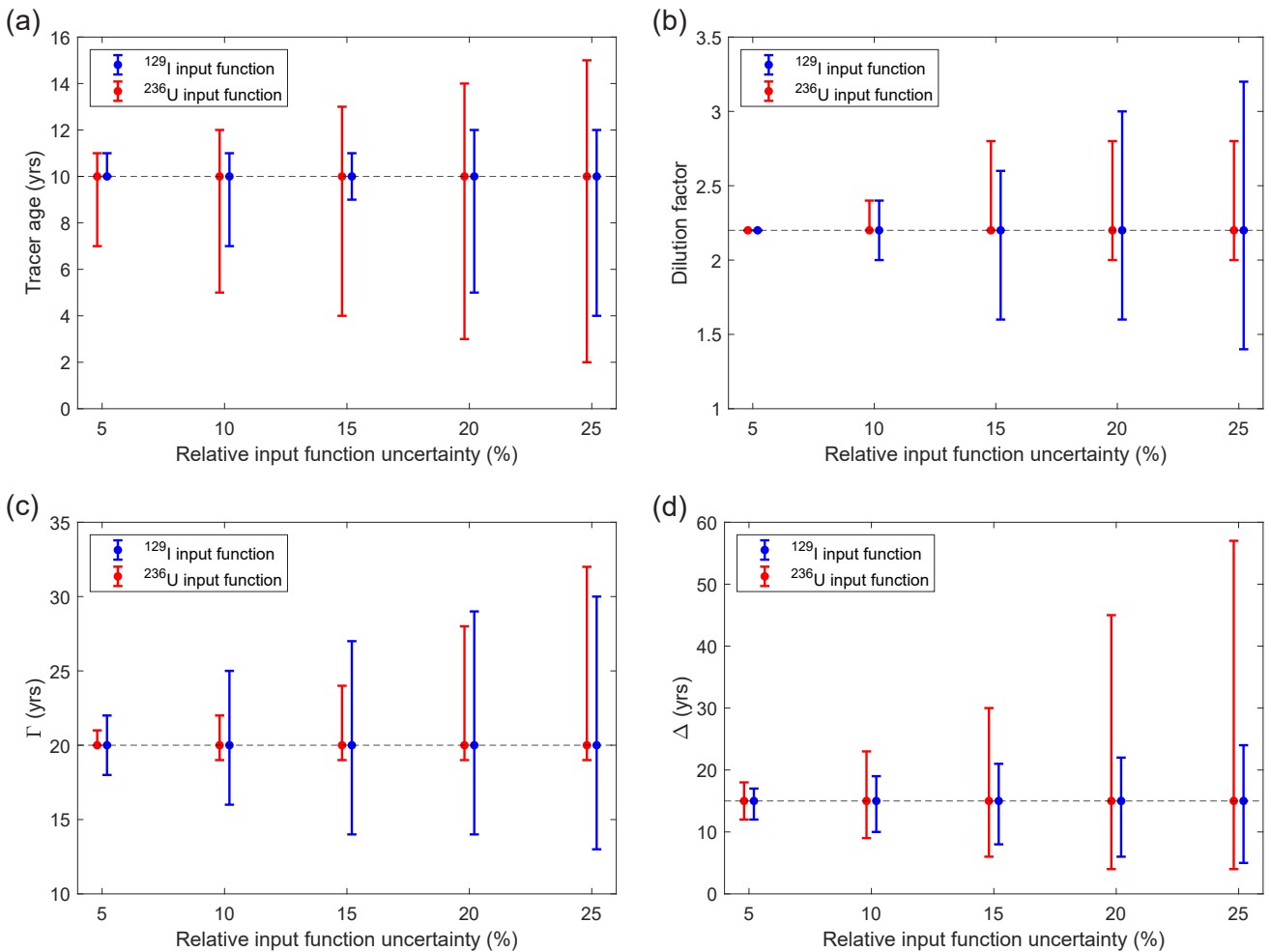

**Figure A1.** Sensitivity analysis for model parameters as a function of uncertainties in the [129]I (blue) and [236]U (red) surface layer input functions. Shown are tracer ages (a) and dilution factors (b) from the binary mixing model, as well as $\Gamma$ (c) and $\Delta$ (d) from the TTD model. Due to the irregular shape of the input functions, the uncertainties depend on the data itself. Here we used the Amundsen Basin mean value as an example. Input functions were shifted by up to 25 % and the model parameters were calculated.

*Author contributions.* A.-M. W. performed the conceptualization, investigation, formal analysis and wrote the original draft. N.C. supported the conceptualization, supervised the study, acquired funding and performed writing – review and editing. M.C. supported the investigation and formal analysis and performed writing – review and editing. N.G. supervised and performed writing – review and editing. J.N.S. supported the conceptualization and performed writing – review and editing.

*Competing interests.* The authors declare no competing interests.

*Acknowledgements.* The authors acknowledge all people involved in the sampling activities across the Arctic Ocean and the subsequent sample processing and measurements, providing the available $^{129}$I and $^{236}$U data across the Arctic Ocean. Special thanks goes to M. López Lora and E. Chamizo Calvo from CNA Sevilla and T. Kenna from the Lamont-Doherty Earth Observatory for sampling and processing $^{236}$U samples from the USCGC Healy cruise in 2015. We also thank G. Henderson from University of Oxford who provided seawater samples
from the ACCACIA cruise in 2013. A.-M. Wefing received funding from the ETH Doctoral Grant ETH-06 16-1 "Combining U-236 with other multi-sourced anthropogenic tracers as a new tool to study ocean circulation". N. Casacuberta's research was supported by the Swiss National Science Foundation (AMBIZIONE PZ00P2_154805) and the Laboratory of Ion Beam Physics at ETH Zurich, which is partially funded by its consortium partners EAWAG, EMPA and PSI. N. Gruber acknowledges support from the European Union's Horizon 2020 research and innovation program under grant agreement No 820989 (COMFORT).
We thank two anonymous reviewers for their comments that contributed to a substantial improvement of this manuscript.

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
