# Peer review of "Circulation timescales of Atlantic Water in the Arctic Ocean determined from anthropogenic radionuclides"

_Ocean Science, 2020_

## Referee Comment (RC1) · Anonymous Referee #1 · 17 Sep 2020

This manuscript uses measurements of two long-lived artificial radionuclides (129I and 236U) together with idealized transport models to analyze the tracer transport in the surface and in the mid-depth Atlantic water in the Arctic Ocean. It is shown that analysis of the pair of radionuclides enables the time scales and pathways of the transport, and role of lateral mixing, to be constrained. The paper is well written, contains interesting new results, and I think is suitable for publication in Ocean Science after only a few relatively-minor revisions (see below).

Specific Comments

1. Why are different models used for surface and mid-depth waters? The manuscript

just states that different models are used, but there needs to be a discussion of why the same model can't be used for both and/or why one model works for one layer and not the other.

2. A related issue is the relationship between the different "ages" reported. Can the tracer ages from the binary model be compared with the mean age (or modal age) from the TTDs? These ages are both shown in summary figure 7, but it needs to be clearer if the difference between the ages reflects real differences in transport times or if some of the differences could be due to the age definition.

3. In the discussion of uncertainties (e.g. appendix A) there should be some mention of the uncertainty in the TTD method due to the choice of G(t). All the analysis assumes the G(t) is an inverse Gaussian, but different mean, and especially mode, ages would be obtained if G had a different functional form. I acknowledge that this uncertainty can't be quantified as easily as some of the other factors, but it should at least be mentioned.

4. The Introduction is very long, with multiple subsections, and it is not until page 5 that what is examined in this paper is discussed. As a reader I greatly prefer papers with more concise Introductions that quickly gets to the outstanding questions and what is examined in the paper. Would it be possible to have a more concise Introduction, with some of the subsection in a second section that provides the background on the radionuclides in the Arctic.

---

## Referee Comment (RC2) · Anonymous Referee #2 · 23 Sep 2020

Wefing et al. present a compilation of I-129 and U-236 measurements collected in the Arctic Ocean over the last 10 years and use this data to determine water circulation timescales using two different models (a binary mixing approach, and the transit time distribution model). The geographic distribution of their data is impressive and allows for a nice comparison of circulation times between the different Arctic basins. I have only minor suggestions, which aim to improve the clarity of the paper and to provide further explanation for some of the statements that have been made in the results & discussion. Comments are listed below, divided by section.

Introduction:

[Figure]

Page 1, line 45: change "...key to the prediction of the nature of..." to "...key to predicting the nature of..."

Page 1, line 46: delete "signals" in the phrase "increased AW temperature signals"

Page 2, Line 28: Pacific waters are not considered part of the Polar Mixed Layer, they reside below this fresher, colder, surface layer. See for example, Figure 15 in Steele et al. 2004, JGR (doi: 10.1029/2003JC002009)

Figure 1: The circulation arrows in the Canada Basin on Figure 1 are not quite correct. The two surface circulation cells in the Canada Basin and Makarov Basin are divided by the Alpha-Mendeleev Ridge. As drawn, the arrows do not follow that boundary (the Makarov Basin arrows should be more parallel to the Amundsen & Nansen circulation arrows). (But nice clear map otherwise!) Be sure to make this fix in Figure 7 as well.

Throughout the paper there is switching between "AW" and "AWs" as notation for Atlantic Water(s). Choose one and keep it consistent to make it easier to understand the text, particularly since there are many acronyms in the paper.

Page 4, line 93: add "us" so the phrase reads "...concentrations of the two tracers then permits us to define..." (alternatively, you could say "concentrations of the two tracers then permits a definition of...")

Page 5, line 108: You say here that there is only one study using the TTD model in the Arctic, but later you compare your results to Tanhua et al. (2009) and Stoeven et al. (2016). Why are those other two studies not mentioned here?

Materials & Methods:

Table 1: How did you choose these depth ranges? Why the gap between surface and Atlantic layer? The Rudels definitions based on density and temperature suggest that the two water masses would be adjacent (i.e. one ends at potential density 27.7, the other begins there).

Page 7, line 155: GF is not defined. I assume it means Global Fallout? I suggest writing out the words rather than introducing another acronym.

Page 7, line 169: RP is not defined. As far as I can tell, this is the only place this acronym is used so I suggest writing out the words.

Page 7, lines 169-172: This explanation of the global fallout signal is repetitive with the previous section. Remove or combine with the previous section rather than repeating here.

Figure 2: In the caption, the phrase "counted from 2019" is confusing because it makes it seem as though 2019 is the year when the count begins rather than ends- it would be better to say "counting back from 2019" or "based on a collection year of 2019". Same for Figure 4 caption.

Page 9, lines 221-223: The explanation of how this parameter relates to lateral mixing should be moved up by lines 200-205, where you first introduce this concept.

Results:

Page 10, line 246: This line says that NCC is carrying high I-129, but in this context I think you mean to say U-236

Figure 3: You make the point that I-129 concentrations are higher in the Atlantic layer compared to the surface in the Canada and Makarov Basins, but this is very hard to see in the figure because they look to be the same shade of blue (even though they are an order of magnitude different in concentration!). Is there a way to scale the colorbar such that this difference can be more clearly seen? Or add a deeper blue color for the very lowest concentrations? Maybe this would make it more difficult to see the changes at higher concentrations, but it is worth testing.

Page 13, line 288: Please explain further what you mean by a change in the circulation pattern (change from what pattern to what pattern?) Could this result also be due to a speeding up of the circulation rather than a shift in geographic location?

Page 13, line 300: Data on the Arctic Oscillation is available- I am familiar with the NOAA data portal (https://psl.noaa.gov/data/climateindices/) but perhaps there are other international portals as well. Based on the observed trends in the AO index, can you confirm that the AO position could produce the observed trends in your data (i.e. was it in a different state in 2012 compared to 2016)? I'm also not sure I fully understand why a shift in the AO position would result in younger ages over the Lomonosov Ridge.

Figure 5: The difference between the dashed and solid lines is hard to discern (particularly for the yellow line). Is it possible to put more space between the dashes to make it clear which line is dashed?

Page 14, line 314: Why is the Atlantic layer considered to be between 250-500 m here compared to the 250-300 m range that is used in other figures/discussion sections? In general it is not clear why each depth range is chosen (aside from the entire Atlantic layer definition given for the 250-800 m range).

Page 15, line 318: Delete "in" so the line reads "determined for the Makarov Basin..."

Page 15: There is no discussion of the one yellow point near the Laptev Shelf in figure 6a. Why does this data point give such an old age?

Discussion:

Page 16, Line 375: Add a reference to Table 2 after the first comparison with the Smith 2011 study.

Page 16, line 388: Pacific water is not part of the polar mixed layer. The point of this paragraph is a good one (that we need more conservative tracers to use in Arctic surface waters), but this should be re-worded to remove the "Polar Mixed Layer" terminology in reference to the Alkire study. They consider Pacific water to be part of the upper halocline in Alkire et al. 2019.

Page 18, lines 403-410: You provide possible explanations for the high mean ages

and significant mixing observed in the Nansen Basin and Fram Strait, but I am curious why this signal also appears near the Laptev Shelf. The mean age estimates for this sample are very different than other estimates in the literature. This large mean age also stands out on Figure 7. Since you do not discuss the Laptev Sea as much as the other basins, I suggest either removing the Laptev Sea box from this figure or expanding on the discussion of this region.

Page 19, line 443-444: This sentence should be re-worded to make it clear that you are still referencing the Mauldin study, for example: "The advective times reported by Mauldlin et al (2010) for the BSBW pathway...". As written, it is not immediately clear which "reported advective times" you are referring to.

Page 19, line 446-447: It is stated that the difference between the mode ages in this study and the Mauldin study may be due to the positive AO phase in the 1990s. Please explain how that would affect the mode ages in the Canada Basin in particular. It is not currently clear how this explains the difference between the two estimates.

Table 3: Explain the acronyms used in the table, or write out the full words (e.g. LS, GL)

Page 21, line 478: Change "improving" to "improve"

Page 22, line 499: Insert "has" so the line reads "...The I and U tracer pair has been shown..."

It is confusing to me that the subscript "max" in tmax (mode age) suggests that this is the maximum age estimate, but the mode age is consistently younger than the mean age. Perhaps it is worth considering a switch to "tmode" or similar.

---

## Author Response (AR1)

Dear Arvind Singh,

I am pleased to submit a revised version of the manuscript "Circulation timescales of Atlantic Waters in the Arctic Ocean determined from anthropogenic radionuclides".

Please find detailed answers addressing the comments of both reviewers below. For specific comments suggesting major changes of certain sections or additional clarifications in the manuscript, the revised sections are cited below the author's response. A marked-up version of the manuscript is included after the response to the reviewers' comments, highlighting all changes made to the originally submitted manuscript.

Please note that we decided to change the title to "Circulation timescales of Atlantic Water in the Arctic Ocean determined from anthropogenic radionuclides" ("Atlantic Water" instead of "Atlantic Waters") to be consistent throughout the whole manuscript.

I would also like to note that I encountered an error in the dataset used for the water mass age calculations during the preparation of the revised version, related to the units of I-129 and U-236 concentrations. The re-calculation of water mass ages with the corrected dataset partly led to slightly different results compared to the originally submitted manuscript. However, as can be seen from the marked-up version of the revised manuscript, these changes are only on the order of a few years and do not change the interpretation or scientific outcome of the study.

I hope that after addressing the reviewers' comments, the revised manuscript will be suitable for publication in Ocean Science.

Yours sincerely,
Anne-Marie Wefing, on behalf of all co-authors

**Point-by-point answer to Referee comment 1 on "Circulation timescales of Atlantic Waters in the Arctic Ocean determined from anthropogenic radionuclides"**
**(from 17 Sep 2020)**

*Dear Reviewer 1,*

*Thank you for your review of the paper and your suggestions on how to improve it. We have incorporated all of them into the revised version of the manuscript. Please find point-by-point answers regarding your comments below. A marked-up version of the revised manuscript can be found after the response to all reviewer comments.*

*On behalf of all co-authors,*
*Anne-Marie Wefing*

This manuscript uses measurements of two long-lived artificial radionuclides (129I and 236U) together with idealized transport models to analyze the tracer transport in the surface and in the mid-depth Atlantic water in the Arctic Ocean. It is shown that analysis of the pair of radionuclides enables the time scales and pathways of the transport, and role of lateral mixing, to be constrained. The paper is well written, contains interesting new results, and I think is suitable for publication in Ocean Science after only a few relatively-minor revisions (see below).

Specific Comments

1. Why are different models used for surface and mid-depth waters? The manuscript just states that different models are used, but there needs to be a discussion of why the same model can't be used for both and/or why one model works for one layer and not the other.

*Reply: Thanks for pointing out that this was not stated clearly enough. We have added some sentences to clarify why the different models were used for the different layers in section 1.3. Please refer to the marked-up version of the manuscript to track the corresponding changes.*

*Concretely, the reason behind using different models for the different layers is the following:*

*Generally, the binary mixing model is only applicable in cases when the advective flow can be assumed to be dominant and lateral mixing within the flow is limited. This assumption is valid only for the surface layer, as has been shown in Smith et al. (2011). In the Atlantic layer, lateral mixing is much more important, invalidating the use of the binary mixing model. In contrast, this process is explicitly accounted for in the TTD model. However, the TTD model does not capture the mixing of different endmembers, e.g., Atlantic and Pacific waters, and therefore cannot be applied to the surface layer.*

2. A related issue is the relationship between the different "ages" reported. Can the tracer ages from the binary model be compared with the mean age (or modal age) from the TTDs? These ages are both shown in summary figure 7, but it needs to be clearer if the difference between the ages reflects real differences in transport times or if some of the differences could be due to the age definition.

*Reply: One should be careful when comparing circulation times between the surface and the Atlantic layers due to the different underlying models and assumptions. In the surface layer, tracer ages from the binary mixing model only reflect actual ages when the AW flow can be considered purely advective. In contrast, the TTD model accounts for mixing and the mean and mode ages are different estimates of circulation times. Only in the case of purely advective flow, mean ages, mode ages, and tracer ages are comparable.*

*However, we found that the mode ages are overall in the same range as the tracer ages and appear to be a suitable measure for the lateral propagation of tracer signals while taking mixing into account.*

*To clarify this in the paper, the last part of section 4.2.2 was changed to:*

> *"In this case, the different age measures deduced from the TTD are also similar and can be compared to tracer ages. If significant mixing is involved, only the TTD model provides meaningful age estimates. We found that overall, mode ages derived from the TTD model for the Atlantic layer are in the same range as tracer ages from the surface layer."*

*In the conclusions, the following sentence was added:*

> *"For a comparison between tracer ages, mean, and mode ages, the different underlying models for the surface and the Atlantic layers need to be considered. In the case of lateral mixing, tracer ages are not meaningful and mean ages will always exceed mode ages. Here, the choice of the age estimate generally depends on the context."*

3. In the discussion of uncertainties (e.g. appendix A) there should be some mention of the uncertainty in the TTD method due to the choice of G(t). All the analysis assumes the G(t) is an inverse Gaussian, but different mean, and especially mode, ages would be obtained if G had a different functional form. I acknowledge that this uncertainty can't be quantified as easily as some of the other factors, but it should at least be mentioned.

*Reply: The choice of G(t) indeed has significant impact on the derived ages. The following comment has been included at the end of section 4.1.3:*

> *"The functional form of the Green's function G(t), describing the propagation of the tracer signal, introduces an uncertainty on the reported mean and mode ages. Examples of different functional forms have for instance been discussed in Haine and Hall (2002). The inverse Gaussian form of G(t), which solves the 1D advection-diffusion equation, has been shown to perform well for many oceanographic applications, and is therefore the most widely used functional form."*

4. The Introduction is very long, with multiple subsections, and it is not until page 5 that what is examined in this paper is discussed. As a reader I greatly prefer papers with more concise Introductions that quickly gets to the outstanding questions and what is examined in the paper. Would it be possible to have a more concise Introduction, with some of the subsection in a second section that provides the background on the radionuclides in the Arctic.

*Reply: We have slightly rearranged the introduction. The following paragraph has been included at the end of section 1.1:*

> *"In this study, we will investigate circulation pathways, mixing regimes as well as tracer ages of AW in the surface and Atlantic layers of the Arctic Ocean. This will be done using a novel approach that combines the two long-lived anthropogenic radionuclides I-129 and U-236. Obtained ages will be put into the context of available literature data and different approaches on how to estimate circulation timescales will be compared. Strengths and weaknesses especially of applying the transit time distribution model introduced in section 1.3 to anthropogenic radionuclides will be discussed and implications for the Arctic Ocean will be highlighted."*

*The last paragraph of the introduction (page 5-6) has consequently been shortened and now reads:*

*"In this study the two models presented above will be applied to revise circulation features and investigate circulation times of AW in the Arctic Ocean using the combination of I-129 and U-236. The detailed methodological approach is outlined in the following section."*

**Point-by-point answer to Referee comment 2 on "Circulation timescales of Atlantic Waters in the Arctic Ocean determined from anthropogenic radionuclides" (from 23 Sep 2020)**

*Dear Reviewer 2,*

*Thank you for your detailed review of the paper. We have incorporated your suggestions into the revised version of the manuscript. Please find point-by-point answers regarding your comments below. A marked-up version of the revised manuscript can be found after the response to all reviewer comments.*

*On behalf of all co-authors,*
*Anne-Marie Wefing*

Wefing et al. present a compilation of I-129 and U-236 measurements collected in the Arctic Ocean over the last 10 years and use this data to determine water circulation timescales using two different models (a binary mixing approach, and the transit time distribution model). The geographic distribution of their data is impressive and allows for a nice comparison of circulation times between the different Arctic basins. I have only minor suggestions, which aim to improve the clarity of the paper and to provide further explanation for some of the statements that have been made in the results & discussion. Comments are listed below, divided by section.

Introduction:

Page 1, line 45: change ". . .key to the prediction of the nature of..." to ". . .key to predicting the nature of. . ."

*Reply: Has been changed.*

Page 1, line 46: delete "signals" in the phrase "increased AW temperature signals"

*Reply: Has been changed.*

Page 2, Line 28: Pacific waters are not considered part of the Polar Mixed Layer, they reside below this fresher, colder, surface layer. See for example, Figure 15 in Steele et al. 2004, JGR (doi: 10.1029/2003JC002009)

*Reply: Thanks for pointing that out. Throughout the paper, we now use the term "surface layer" or "Polar Surface Water" (which comprises the Polar Mixed Layer and the halocline, Rudels et al., 2005) instead of Polar Mixed Layer.*

*Page 2, line 28 was changed to:*

> *"Both Pacific Water and freshwater largely reside in the upper water column of the Arctic Ocean, including the Polar Mixed Layer and the upper halocline."*

Figure 1: The circulation arrows in the Canada Basin on Figure 1 are not quite correct. The two surface circulation cells in the Canada Basin and Makarov Basin are divided by the Alpha-Mendeleev Ridge. As drawn, the arrows do not follow that boundary (the Makarov Basin arrows should be more parallel to the Amundsen & Nansen circulation arrows). (But nice clear map otherwise!) Be sure to make this fix in Figure 7 as well.

*Reply: Thanks, both Fig. 1 and Fig. 7 have been changed accordingly.*

Throughout the paper there is switching between "AW" and "AWs" as notation for Atlantic Water(s). Choose one and keep it consistent to make it easier to understand the text, particularly since there are many acronyms in the paper.

*Reply: We decided to use "AW" only and changed the paper accordingly. In the abstract, we now only use the term "Atlantic Water".*

Page 4, line 93: add "us" so the phrase reads ". . .concentrations of the two tracers then permits us to define. . ." (alternatively, you could say "concentrations of the two tracers then permits a definition of. . .")

*Reply: Has been changed.*

Page 5, line 108: You say here that there is only one study using the TTD model in the Arctic, but later you compare your results to Tanhua et al. (2009) and Stoeven et al. (2016). Why are those other two studies not mentioned here?

*Reply: On page 5, line 108, we specifically refer to the use of the TTD model to determine lateral transit times of Atlantic Waters rather than ventilation times, by using anthropogenic radionuclides released by nuclear reprocessing plants. In this regard, (to our knowledge) only the study by Smith et al. (2011) was conducted. This is now clarified in the paper. The studies by Tanhua et al. (2009) and Stöven et al. (2016) are mentioned some paragraphs above, in the context of other ventilation studies, when introducing the TTD model in general (page 4, lines 85-88).*

*The paragraph now reads:*

> *"To our knowledge only one study has applied the TTD model to determine lateral transit times of AW in the Arctic Ocean using anthropogenic radionuclides (Smith et al., 2011). These authors combined radionuclides from nuclear reprocessing (129I and 137Cs) with the atmospherically introduced tracer CFC-11."*

Materials & Methods:

Table 1: How did you choose these depth ranges? Why the gap between surface and Atlantic layer? The Rudels definitions based on density and temperature suggest that the two water masses would be adjacent (i.e. one ends at potential density 27.7, the other begins there).

*Reply: We defined only two input functions for radionuclides entering the Arctic Ocean, one in the Norwegian Coastal Current and one for the FSBW and BSBW. Therefore, we chose to limit the application of the binary mixing model to the surface mixed layer and the TTD model to the mid-depth Atlantic layer. Table 1 was slightly changed, now including an extra column for the depth ranges (see marked-up manuscript).*

Page 7, line 155: GF is not defined. I assume it means Global Fallout? I suggest writing out the words rather than introducing another acronym.

*Reply: Has been changed to "global fallout" throughout the paper.*

Page 7, line 169: RP is not defined. As far as I can tell, this is the only place this acronym is used so I suggest writing out the words.

*Reply: Has been changed to "reprocessing plant".*

Page 7, lines 169-172: This explanation of the global fallout signal is repetitive with the previous section. Remove or combine with the previous section rather than repeating here.

*Reply: The first paragraph of section 2.3.1 has been changed and now reads*

> *"In our application of the binary mixing model for the surface layer of the Arctic Ocean, we assumed that water labeled with the time-dependent tracer signal entering the Arctic Ocean in the surface AW input function only mixes with water carrying the global fallout signal of 1x10^7 at I^(-1) for I-129 and U-236 (i.e., Pacific and Atlantic water not labeled with reprocessing plant releases)."*

Figure 2: In the caption, the phrase "counted from 2019" is confusing because it makes it seem as though 2019 is the year when the count begins rather than ends- it would be better to say "counting back from 2019" or "based on a collection year of 2019". Same for Figure 4 caption.

*Reply: These phrases have been changed to "based on a collection year of 2019" (caption of Fig. 2) and "based on a collection year of 2015" (caption of Fig. 4).*

Page 9, lines 221-223: The explanation of how this parameter relates to lateral mixing should be moved up by lines 200-205, where you first introduce this concept.

*Reply: The explanation of Delta has been moved up to the description of Gamma and Delta, which now reads*

> *"Delta is related to the second moment of the PDF and is a measure of its width, i.e., it describes how much a tracer signal disperses during the flow as a result of lateral mixing. The larger this mixing, the more incorporation of waters with different ages occurs within a water parcel sampled downstream from the initialization point of the input function. For Delta = 0, …"*

Results:

Page 10, line 246: This line says that NCC is carrying high I-129, but in this context I think you mean to say U-236

*Reply: True, "I-129" has been changed to "U-236".*

Figure 3: You make the point that I-129 concentrations are higher in the Atlantic layer compared to the surface in the Canada and Makarov Basins, but this is very hard to see in the figure because they look to be the same shade of blue (even though they are an order of magnitude different in concentration!). Is there a way to scale the colorbar such that this difference can be more clearly seen? Or add a deeper blue color for the very lowest concentrations? Maybe this would make it more difficult to see the changes at higher concentrations, but it is worth testing.

*Reply: The scale of the I-129 colorbar in Figure 3 has been adjusted to be able to distinguish better between low I-129 concentrations (see revised manuscript). Thanks for your suggestions.*

Page 13, line 288: Please explain further what you mean by a change in the circulation pattern (change from what pattern to what pattern?) Could this result also be due to a speeding up of the circulation rather than a shift in geographic location?

*Reply: See answer to the comment below (addressing page 13, line 300). This sentence was moved to the paragraph below, discussing the changes between sampling years. In*

*principle this change could be due to a speeding up of the circulation but this is highly speculative based only on data from two years.*

Page 13, line 300: Data on the Arctic Oscillation is available- I am familiar with the NOAA data portal (https://psl.noaa.gov/data/climateindices/) but perhaps there are other international portals as well. Based on the observed trends in the AO index, can you confirm that the AO position could produce the observed trends in your data (i.e. was it in a different state in 2012 compared to 2016)? I'm also not sure I fully understand why a shift in the AO position would result in younger ages over the Lomonosov Ridge.

*Reply: Between 2011/12 and 2015, no clear trend in the AO index could be observed. However, a shift in the AO index could result, in general, in a change of the spreading of Atlantic waters across the Arctic Ocean, especially in the Makarov Basin. A change in the fraction of Atlantic-derived waters at a certain sampling location mainly influences the dilution factors obtained from the binary mixing model. For instance, more Pacific Waters present at the stations in the Makarov Basin would lead to higher dilution factors, as observed for 2015.*

*The tracer ages obtained from the binary mixing model should not directly be affected by a higher Pacific Water fraction bringing the global fallout signal, as this should only move the samples towards to global fallout endmember along the binary mixing line. However, the mixing lines converge towards the global fallout endmember and hence the tracer age determination becomes less precise. In addition, the change in tracer ages could reflect changing AW pathways resulting from a shift in the location of the Atlantic-Pacific Water front.*

*The paragraph about changes observed between sampling years for tracer ages and dilution factors has been changed and now reads:*

> *"Regarding differences between sampling years (2011/12 vs. 2015), a slight decrease of tracer ages over time could be observed for the central Arctic Ocean (Fig. 4b). In the Makarov Basin, dilution factors significantly increased from about 2 in2011/12 to 10-20 in 2015. This implies that the Pacific Water fraction increased over time and can be explained by a shift of the Atlantic-Pacific water front that generally determines the water mass provenance in the Makarov Basin.*

> *The alignment of the Atlantic-Pacific front across the Arctic Ocean is influenced by the prevailing atmospheric circulation (indicated by, e.g., the Arctic Oscillation index; Morison et al. (2012); Alkire et al. (2019) and ref. therein). However, no clear trend in the AO index was observed between 2011/12 and 2015 and hence the reason behind changes in the Pacific Water fraction in the Makarov Basin remains unclear. Tracer ages should not be affected by the water mass provenance but could change as a result of potentially changing AW pathways in the central Arctic Ocean."*

Figure 5: The difference between the dashed and solid lines is hard to discern (particularly for the yellow line). Is it possible to put more space between the dashes to make it clear which line is dashed?

*Reply: For a better visibility, the Amundsen Basin data points in Fig. 4a and 5a, as well as the IG-TTD line (Fig. 5b) have been changed to magenta instead of yellow.*

Page 14, line 314: Why is the Atlantic layer considered to be between 250-500 m here compared to the 250-300 m range that is used in other figures/discussion sections? In

general it is not clear why each depth range is chosen (aside from the entire Atlantic layer definition given for the 250-800 m range).

*Reply: We agree, the choice of depth layers was not entirely consistent. Figure 5a has been changed such that all samples between 250 and 800m depth are averaged and plotted here.*

Page 15, line 318: Delete "in" so the line reads "determined for the Makarov Basin. . ."

*Reply: Has been changed.*

Page 15: There is no discussion of the one yellow point near the Laptev Shelf in figure 6a. Why does this data point give such an old age?

*Reply: A possible explanation for the high mean age and Delta/Gamma ratio of this sample could be the choice of the input function. The following sentences have been included in the discussion of the mean ages (page 15):*

> *"A high mean age and the highest Delta/Gamma ratio were found for one station close to the Laptev Sea shelf. This result was rather unexpected given the proximity of this station to the initialization point of the input functions and was due to the comparably low I129 and U236 concentrations measured at 300m depth. A possible explanation could be a higher influence of the FSBW input function at that depth, rather than the mixture of FSBW and BSBW that is used for our Atlantic layer input function. Using the FSBW input function instead, the mean age for this station would be on the order of 10 years and the Delta/Gamma ratio would be about 0.5, which is much more reasonable. In the context of this study, the Laptev Sea shelf data point will therefore not be discussed further."*

Discussion:

Page 16, Line 375: Add a reference to Table 2 after the first comparison with the Smith 2011 study.

*Reply: The reference to Table 2 has been added.*

Page 16, line 388: Pacific water is not part of the polar mixed layer. The point of this paragraph is a good one (that we need more conservative tracers to use in Arctic surface waters), but this should be re-worded to remove the "Polar Mixed Layer" terminology in reference to the Alkire study. They consider Pacific water to be part of the upper halocline in Alkire et al. 2019.

*Reply: see reply to comment above (addressing page 2, line 28). "Polar Mixed Layer" has been changed to surface layer or Polar Surface Water throughout the paper.*

Page 18, lines 403-410: You provide possible explanations for the high mean ages and significant mixing observed in the Nansen Basin and Fram Strait, but I am curious why this signal also appears near the Laptev Shelf. The mean age estimates for this sample are very different than other estimates in the literature. This large mean age also stands out on Figure 7. Since you do not discuss the Laptev Sea as much as the other basins, I suggest either removing the Laptev Sea box from this figure or expanding on the discussion of this region.

*Reply: See answer to the comment above (adressing page 16, line 388). The Laptev Sea box was removed from Figure 7.*

Page 19, line 443-444: This sentence should be re-worded to make it clear that you are still referencing the Mauldin study, for example: "The advective times reported by Mauldin et al

(2010) for the BSBW pathway. . .". As written, it is not immediately clear which "reported advective times" you are referring to.

*Reply: The sentence has been changed to "The advective times reported by Mauldin et al. (2010) for the BSBW pathway…"*

Page 19, line 446-447: It is stated that the difference between the mode ages in this study and the Mauldin study may be due to the positive AO phase in the 1990s. Please explain how that would affect the mode ages in the Canada Basin in particular. It is not currently clear how this explains the difference between the two estimates.

*Reply: Mauldin et al. (2010) report advective times for transport within the Arctic Ocean Boundary Current (AOBC). The positive AO phase during the 1990s can be associated to accelerated boundary current flow (as described in Karcher et al. 2012) and therefore low advective ages in the Canada Basin. The mode age obtained from I-129 and U-236 for 2015 cannot clearly be attributed to the boundary current and the pathways of Atlantic waters to the Canada Basin probably differ as consequence of a weakening of the AOBC due to the transition of the AO to a more negative phase (Karcher et al. 2012). Since we have only one station covering the Atlantic layer in the Canada Basin we cannot state how the circulation modes affect the mode ages. However, there is currently a new paper under review in Journal of Geophysical Research: Oceans which examines the dispersion of I-129 in more detail, under different AO phases (pers. comm. J.N. Smith).*

*This explanation has been included in the corresponding paragraph which now reads:*

> *"Here it should be noted that the latter employed data from the 1990s when the Arctic Oscillation was in a positive phase and exceptionally strong cyclonic boundary current conditions prevailed in the Arctic Ocean, associated with accelerated boundary current flow (Karcher et al., 2012). This implies low advective ages. The Canada Basin mode age obtained from I-129 and U-236 for 2015 cannot unambiguously be attributed to the boundary current. Additionally, AW pathways to the Canada Basin probably changed as a consequence of a weakening of the boundary current due to the transition of the AO to a more negative phase (Karcher et al., 2012). "*

Table 3: Explain the acronyms used in the table, or write out the full words (e.g. LS, GL)

*Reply: All acronyms in Table 2 and 3 that are not used throughout the paper have been written out in full words (GL – Greenland, LS – Laptev Sea, NP - North Pole).*

Page 21, line 478: Change "improving" to "improve"

*Reply: Has been changed.*

Page 22, line 499: Insert "has" so the line reads ". . .The I and U tracer pair has been

shown. . ."

*Reply: Has been changed.*

It is confusing to me that the subscript "max" in tmax (mode age) suggests that this is the maximum age estimate, but the mode age is consistently younger than the mean age. Perhaps it is worth considering a switch to "tmode" or similar.

*Reply: This might indeed be confusing, "$t_{max}$" was changed to "$t_{mode}$" throughout the paper.*

[revised manuscript text omitted]

---

## Author Response (AR2)

Dear Arvind Singh,

I am pleased to submit the final version of the manuscript "Circulation timescales of Atlantic Water in the Arctic Ocean determined from anthropogenic radionuclides".

I would like to address the minor comment of referee #2:

"As I pointed out in my first review, the depth range chosen for the Atlantic layer is different between sections 3.1 (Figure 3) and 3.3 (Figure 5). The authors changed the range in section 3.3 to be consistent with the Atlantic Water definition give in Table 1, which is helpful, but they did not change it in section 3.1, or in Figures 5 and 6. If the goal is to discuss different parts of the Atlantic layer (e.g. FSBW vs. BSBW), it would be helpful to define this in Table 1 rather than defining it as one whole layer."

*Reply: The depth range in Fig. 3 was chosen to give an overview on I-129 and U-236 concentrations in the core of the Atlantic layer, which we here considered to be represented by samples between 250 and 300m depth. In the surface plot, it is only reasonable to show samples from a certain depth layer, rather than including the whole water column of the Atlantic layer. This also holds true for Fig. 6. Therefore, also the discussion of Fig. 3 (section 3.1) and Fig. 6 (second part of section 3.3) focuses on the same depth layer (the depth range referred to on p.10, line 256 should actually read "250-300m", this has been changed). In both cases, the distribution of results across the Arctic Ocean is described.*

*In contrast, the objective of Fig. 5 is to give an overview where the different Arctic basins plot in the domain of Gamma and Delta isolines of the TTD model hence samples from the entire Atlantic layer were included. The depth range in the caption of Fig. 5 was corrected to "250-800m".*

I hope this response sufficiently addresses the comment of referee #2.

Yours sincerely,
Anne-Marie Wefing, on behalf of all co-authors